# MILEBENCH: Benchmarking MLLMs in Long Context

**Dingjie Song, Shunian Chen, Guiming Hardy Chen, Fei Yu, Xiang Wan, Benyou Wang**[*]
The Chinese University of Hong Kong, Shenzhen
Shenzhen Research Institute of Big Data
wangbenyou@cuhk.edu.cn
https://milebench.github.io/

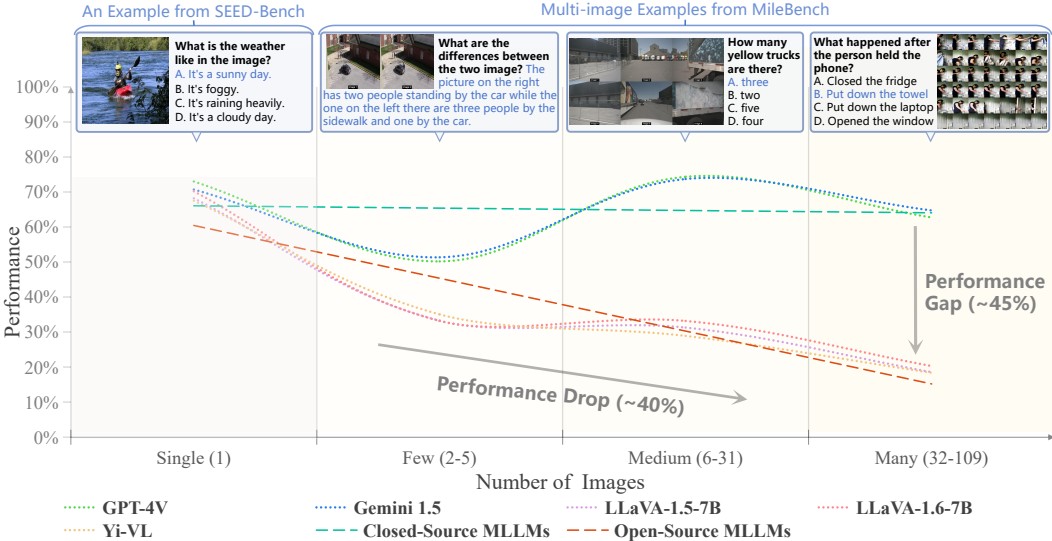

Figure 1: MLLMs' performance fluctuates with the image count in datasets. Open-source MLLMs demonstrate a remarkable performance drop as the number of images increases. The performance gap between open-source and closed-source MLLMs expands as well. For single-image performance, we refer to SEED-Bench (Li et al., 2023b), given the absence of single-image samples in MILEBENCH.

## Abstract

Despite the advancements and impressive performance of Multimodal Large Language Models (MLLMs) on benchmarks, their effectiveness in real-world, long-context, and multi-image tasks is unclear due to the benchmarks' limited scope. Existing benchmarks often focus on single-image and short-text samples, and when assessing multi-image tasks, they either limit the image count or focus on specific task (e.g time-series captioning), potentially obscuring the performance challenges of MLLMs. To address these limitations, we introduce **MILEBENCH**, a pioneering benchmark designed to test the **M**ult**I**modal **L**ong-cont**E**xt capabilities of MLLMs. This benchmark comprises not only multimodal long contexts, but also multiple tasks requiring both comprehension and generation. We establish two distinct evaluation sets, diagnostic and realistic, to systematically assess MLLMs' long-context adaptation capacity and their ability to complete tasks in long-context scenarios. Our experimental results, obtained from testing 22 models, revealed that while the closed-source GPT-4o outperforms others, most open-source MLLMs struggle in long-context situations. Interestingly, the performance gap tends to widen with an increase in the number of images. We strongly encourage an intensification of research efforts towards enhancing MLLMs' long-context capabilities, especially in scenarios involving multiple images.

---

[*]Corresponding author.

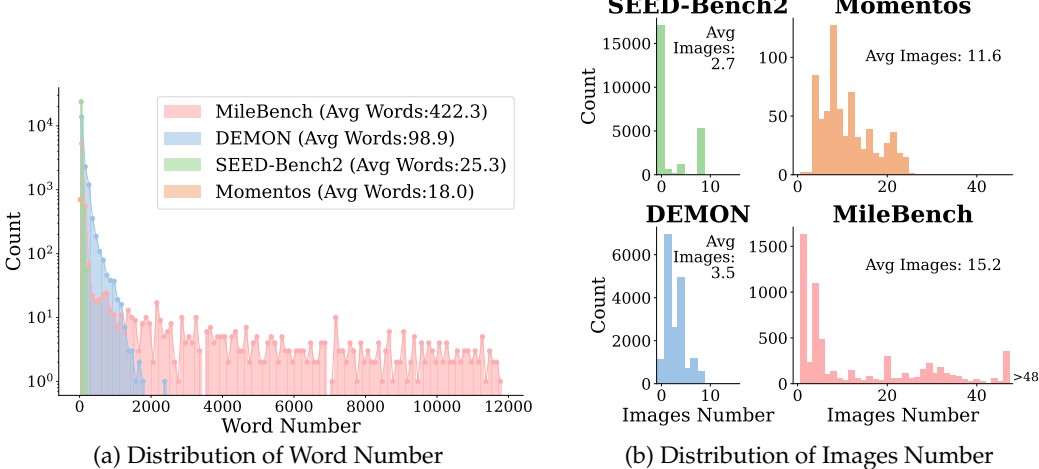

(a) Distribution of Word Number      (b) Distribution of Images Number

Figure 2: Visualization of distribution of images and word number in present MLLM benchmarks. The range and mean of both word and image number per sample in MILEBENCH far exceed those of previous works.

# 1 Introduction

The recent swift development of Multimodal Large Language Models (MLLMs) (OpenAI, 2023; Anil et al., 2023; Liu et al., 2023a; Awadalla et al., 2023) has displayed outstanding performance across a diverse range of multimodal tasks (Yang et al., 2023; Wu et al., 2023). Meanwhile, a surge of benchmarks for evaluating MLLM performance has emerged (Liu et al., 2023a; Fu et al., 2023; Ge et al., 2023; Li et al., 2023a), offering insights into their general capabilities (Li et al., 2023b; Liu et al., 2023c; Yu et al., 2023) and task-specific capabilities (Liu et al., 2023d; Yue et al., 2023; Wang et al., 2024).

However, a critical aspect is often overlooked. Real-world applications frequently demand the processing of long contexts and multi-image tasks that include multi-round dialogues based on multiple images (Li et al., 2022), action prediction tasks (Wu et al., 2021), navigation tasks in 3D space (Krantz et al., 2020), and understanding tasks with lengthy documents interspersed with images on Wiki pages (Hannan et al., 2020).

Despite this, **existing benchmarks primarily focus on single-image and short-text samples** (Liu et al., 2023a; Fu et al., 2023; Liu et al., 2023c; Li et al., 2023b), thereby failing to fully capture the complexity and diversity of real-world scenarios. While some benchmarks evaluate multi-image tasks, they either have limited number of images provided per sample (e.g., SEED-Bench-2 (Li et al., 2023a), DEMON (Li et al., 2023c)) or only include time-series captioning tasks (e.g., Mementos (Wang et al., 2024)), as shown in Figure 2. In addition, this omission could potentially neglect the hallucination issue that MLLMs might exhibit in long-context situations (Huang et al., 2023). Given the aforementioned shortcomings with existing benchmarks, we identify a pressing need for a more holistic evaluation that fully encapsulates the long-context and multi-image task demands prevalent in real-world applications.

Addressing this need, we introduce **MILEBENCH**, the first benchmark specifically designed to test the **M**ult**I**modal **L**ong-cont**E**xt capabilities of MLLMs[1]. To systematically assess the capabilities of MLLM in multimodal long contexts, our benchmark consists of two distinct evaluation sets, **diagnostic evaluation** and **realistic evaluation**. The former explores the long-context recall abilities of MLLMs, using needle-in-a-haystack and image retrieval tasks, while the latter stress-tests the model in a manner akin to real-world conditions using both temporal multi-image tasks and semantic multi-image tasks. To construct our evaluation sets, we gather 6,440 multimodal long-context samples from 21 pre-existing or self-constructed datasets, with an average of 15.2 images and 422.3 words each, as depicted in Figure 2, and we categorize them into their respective subsets.

---

[1]We define "multimodal long contexts" as long text content integrated with two or more images, or content composed of multiple images.

After evaluating 22 models, the closed-source GPT-4o[2] excelled in both diagnostic and realistic evaluations, achieving impressive scores of 99.4% and 60.3%, although it still falls short of a perfect 100% score. On the contrary, most open-source MLLMs struggled with long-context tasks as depicted in Figure 1. Only Mantis and Qwen-VL-7B managed average scores of 47.5% and 37.2% in realistic and diagnostic evaluations respectively. These results underscore that there are **"miles to go"** towards fully-realized long-context MLLMs, prompting a call for increased research focus on such tasks, especially those involving numerous images.

## 2 Related Work

### 2.1 Multi-image and Long-Context MLLMs

Beyond training on single-image-text pairs, recent developments in MLLMs are also oriented towards handling multiple and interleaved image-text sequences (Awadalla et al., 2023; Li et al., 2023c). However, these models have relatively limited contexts (i.e., up to 4K) compared to leading proprietary MLLMs such as GPT-4V (OpenAI, 2023) and Gemini (Anil et al., 2023; Reid et al., 2024), which exhibit capabilities for long-context processing, supporting up to 128K and 10M tokens, respectively. However, there remains a notable gap in open-source MLLMs capable of long-context comprehension. Currently, the only open-source models equipped for long contexts are those designed for video, which are trained to process multiple frames, inherently managing multiple images and long contexts (Liu et al., 2024a; Zhang et al., 2023; Luo et al., 2023; Li et al., 2023d;e). In this paper, we release an evaluation set specifically designed for multi-image and long-context MLLMs.

### 2.2 Evaluation of MLLMs

Most of the MLLM benchmarks only evaluate multimodal tasks with a single image (Liu et al., 2023a; Fu et al., 2023; Liu et al., 2023c; Li et al., 2023b; Yu et al., 2023; Ge et al., 2023; Yue et al., 2023). As a complementarity, SEED-Bench2 (Li et al., 2023a) and DEMON (Li et al., 2023c) test multimodal capabilities with multiple images but limit the evaluation to around three images, which is inadequate for a thorough multi-image comprehension assessment. Mementos (Wang et al., 2024) involves data samples with up to approximately 11 images, mainly focusing on temporal understanding and limited context scenarios. This focus overlooks the wide range of scenarios involving multiple images and long context. We provide an overview of existing MLLM benchmarks in Appendix A. To the best of our knowledge, MILEBENCH is the first comprehensive benchmark that evaluates MLLMs across both multi-image and long-context dimensions, catering to a broader spectrum of general scenarios.

## 3 MILEBENCH

### 3.1 Evaluation Taxonomy

MILEBENCH consists of two major components: **Realistic Evaluation** and **Diagnostic Evaluation**, as depicted in Figure 3. **Realistic Evaluation** requires MLLMs to address tasks within multimodal long-context scenarios, emphasizing the models' proficiency in comprehending and reasoning across extended multimodal contexts. Conversely, **Diagnostic Evaluation** demands MLLMs to retrieve information from the provided context, highlighting the model's capability of long-range information retrieval and the elimination of distractors. The detailed taxonomy of MILEBENCH is illustrated in Table 4. [3]

#### 3.1.1 Realistic Evaluation

The realistic evaluation is designed to assess an MLLM's ability to comprehend, integrate, and infer information in a multimodal long context. We categorize the tasks into two main groups: **Temporal Multi-Image tasks** and **Semantic Multi-Image tasks**. Temporal

---

[2]https://openai.com/index/hello-gpt-4o

[3]We adapted the taxonomy from MVBench (Li et al., 2024) and DEMON (Li et al., 2023c) to suit our multimodal long-context setting.

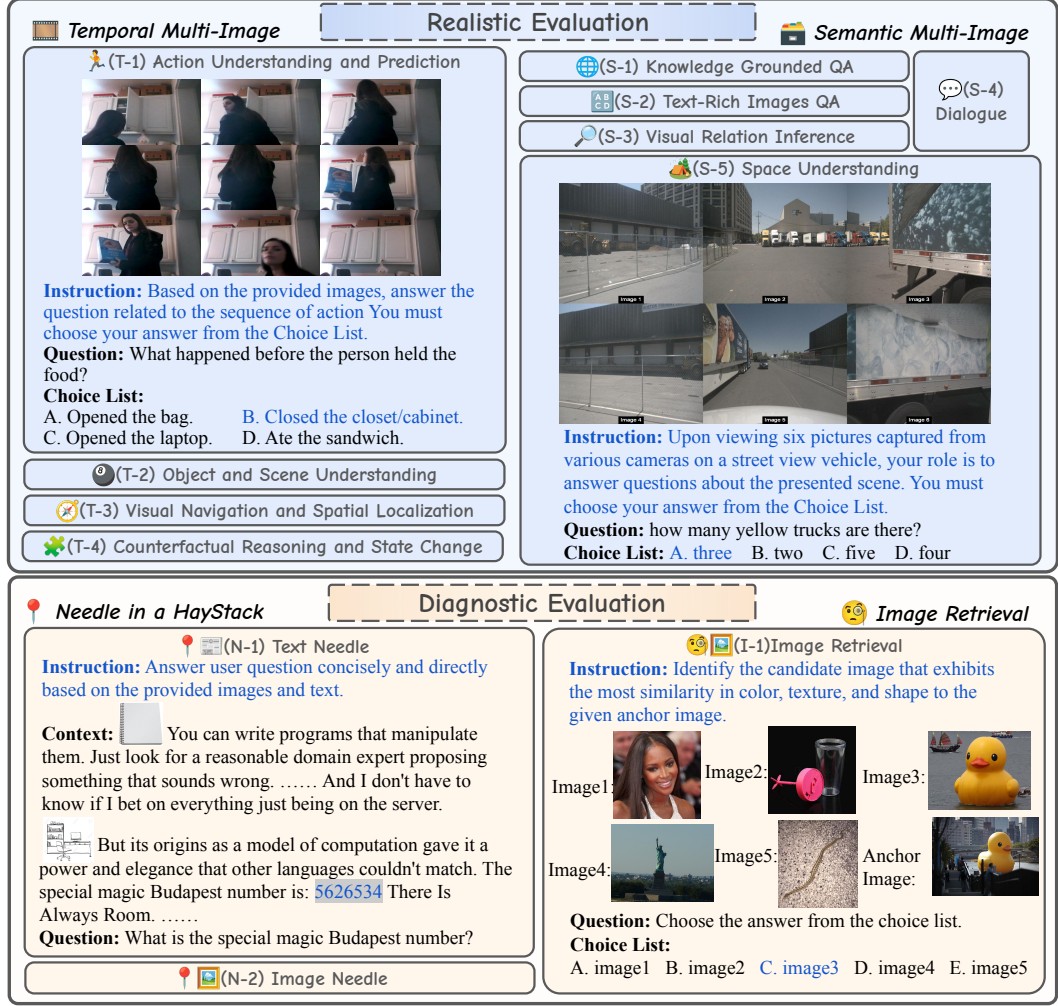

Figure 3: Taxonomy and four multimodal long-context examples in MILEBENCH.

Multi-Image tasks test the MLLM's ability to discern temporal relationships among several time-related images, emphasizing the model's predictive capabilities in real-world scenarios. On the other hand, Semantic Multi-Image tasks challenge MLLMs to process multiple images that are possibly temporal-irrelevant but are semantically interconnected.

**Temporal Multi-Image Tasks.** Temporal Multi-Image tasks include four temporal-related multi-image tasks. Each task contains multiple subtasks, each with 200 samples.

- T-1 [4] **Action Understanding and Prediction** task involves interpreting and forecasting actions of objects or characters in sequential scenarios based on a series of images. This task is divided into three subtasks. *Action Localization* (Gao et al., 2017) assesses the model's ability to identify time actions within a sequence. *Action Prediction* (Wu et al., 2021) tests the model's capacity to predict a character's actions. *Action Sequence* (Wu et al., 2021) evaluates the model's understanding of the chronological order of a character's actions.

- T-2 **Object and Scene Understanding** task involves identifying and understanding objects within a sequence. It comprises four subtasks: *Object Existence* (Yi et al., 2020) tests the model's ability to detect and track an object's movement. *Moving Attribute* (Yi et al., 2020) assesses the model's understanding of moving object attributes. *Object Interaction* (STAR) evaluates the model's comprehension of inter-

---

[4] **T**, **S**, **N**, **I** denotes **T**emporal Multi-image Tasks, **S**emantic Multi-image Tasks, **N**eedle in a Haystack Tasks, and **I**mage Retrieval Tasks, respectively. The numerical value is the index of the task within the set of tasks.

actions between people and objects in complex scenarios. *Object Shuffle* (Patraucean et al., 2023) gauges the model's ability to locate hidden objects amidst disturbances.

T-3 **Visual Navigation and Spatial Localization** task tests the model's understanding of spatial and directional concepts through two subtasks. One is *Egocentric Navigation* (Krantz et al., 2020), involves the model interpreting motion-related instructions and image sequences from a robot's perspective to predict the next action. *Moving Direction* (Yi et al., 2020), assesses the model's ability to determine object movement direction, evaluating its understanding of spatial orientation.

T-4 **Counterfactual Reasoning and State Change** task evaluates the model's logical reasoning within image sequences, focusing on causality and state changes. It comprises four subtasks. *Counterfactual Inference* (Yi et al., 2020) tests the model's ability to predict outcomes under hypothetical changes. *State Change* (Patraucean et al., 2023) assesses the model's understanding of object state changes. *Character Order* (Patraucean et al., 2023) examines the model's reasoning of the order of letter appearances over time. *Scene Transition* (Huang et al., 2020) evaluates the model's understanding of scene changes and the associated causality.

**Semantic Multi-Image Tasks.** Semantic Multi-Image tasks include five semantic-related multi-image tasks. Each task contains multiple existing or artificially constructed datasets, each with 200 samples.

S-1 **Knowledge Grounded QA** task centres on knowledge-based reasoning, where models synthesise multimodal knowledge for single- or multi-hop reasoning tasks. The task employs four datasets: *Webpage QA* (Chang et al., 2022) for open-domain multi-hop web search, *Textbook QA* (Kembhavi et al., 2017) for multimodal textbook questions with diagrams and images, *Complex Multimodal QA* (Talmor et al., 2021) for complex Wikipedia questions with tables and images, and *Long Text with Images QA* for long text with images questions from Wiki documents.

S-2 **Text-Rich Images QA** task demands the processing and understanding of rich text information embedded directly in images. It calls for models capable of recognizing textual information from images and integrating this textual information with complex reasoning to answer questions. It contains *Slide QA* (Tanaka et al., 2023) for multi-slides question answering requiring multi-hop and numerical reasoning, *OCR QA* (Mishra et al., 2019) for book cover image text reading, and *Document QA* (Mathew et al., 2021) for document images with a focus on understanding document structure.

S-3 **Visual Relation Inference** task is centered around understanding and inferring visual relationships. It aims to detect subtle variations between two images, such as changes in objects and positional shifts, and subsequently generate accurate descriptions of these changes. This necessitates the model to possess robust capabilities in capturing visual details and generating response. The task leverages *Visual Change Captioning* (Jhamtani & Berg-Kirkpatrick, 2018; Hosseinzadeh & Wang, 2021) to describe differences between similar images, image change capturing, and *Visual Relationship Expressing* (Tan et al., 2019) for generating relationship captions between images.

S-4 **Dialogue** task primarily involve the fusion of visual data and natural language dialogue understanding. This task needs the model to understand and process multimodal data, including both textual and visual information, while demonstrating consistency and complementarity in the task. The datasets involve *Multimodal Dialogue* (Li et al., 2022) for multimodal conversational question answering and *Conversational Embodied Dialogue* (Shridhar et al., 2020) for mapping from language commands and visuals to action sequences.

S-5 **Space Understanding** task requires the model to perceive the spatial environment using multi-image information. It uses the *nuscenes* (Caesar et al., 2020) dataset, specifically designed for self-driving car technology. It contains sensor data for object detection and tracking, with 1000 annotated scenes for location and properties of objects.

Table 1: **Key Statistics of MILEBENCH.** Note that we use tokenizer of LLaMA2 to calculate the token number.

| Statistic | Number |
|---|---|
| Total samples | 6,440 |
| Total images | 97,855 |
| **Average images** | **15.2** |
| **Range of images** | **2~109** |
| **Range of words** | **7~11821** |
| (Estimated) Average Tokens | |
| - Image token=0 | 542.2 |
| - Image token=32 | 1,028.4 |
| - Image token=256 | 4,432.1 |
| - Image token=576 | 9,294.4 |
| Samples by Image Num Level | |
| - Few (2~5) | 2,959 |
| - Medium (6~31) | 2,389 |
| - Many (32~109) | 1,092 |

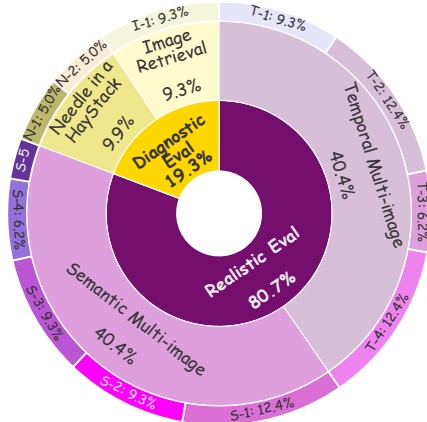

Figure 4: **Tasks Distribution of MILEBENCH.**

### 3.1.2 Diagnostic Evaluation

The diagnostic evaluation focuses on the MLLMs' capability to retrieve information without being distracted in a multimodal long context. We transform the tasks of "Needle in a Haystack" from NLP and "Image Retrieval" from CV into a multimodal format for assessment. This transition preserves the core of the conventional tasks while offering a more challenging and realistic measure of MLLMs' performance.

**Needle in a Haystack.** The "Needle in a Haystack" task requires the model to find a preset password from a long context. This is widely used in diagnostic evaluations of long-context language models (Kuratov et al., 2024). In this study, we reintroduce this novel task from a multimodal perspective to evaluate the context perceptual and retrieval abilities of MLLMs.

Specifically, we constructed two tasks, *Text Needle In A Haystack* (N-1) and *Image Needle In A Haystack* (N-2). Examples are shown in the lower left corner of Figure 3 and Figure 9 in Appendix B.2, respectively. Compared to unimodal Needle in a Haystack task, *Text Needle In A Haystack*'s haystack includes both text and images and the model is required to recall a randomly generated 7-digit password from this multimodal haystack. *Image Needle In A Haystack* changes the modality of the "needle", inserting it as text within an image. This cross-modal Needle in a Haystack task requires MLLMs to have not only good perceptual and retrieval abilities but also robust OCR capabilities.

**Image Retrieval.** Image Retrieval task (I-1) (Schall et al., 2022) requires the model to retrieve images from a set of candidates images given an anchor image (an example is shown in the lower right corner of Figure 3). In addition to perceptual and retrieval abilities, this task also necessitates that the MLLM possesses robust abilities in both object and conceptual recognition. We consider this a traditional computer vision task and see image retrieval as a "Needle in a Haystack" task with image modality queries.

### 3.2 Data Collection and Review Process

We have established a robust data collection process and meticulous review procedures to maintain the integrity and quality of our datasets.

**Data Collection.** We collected the samples from two sources: (1) For most of the tasks, we selected and sampled 200 instances from the test sets of the pre-existing datasets for each task, giving priority to multi-image samples. For video data, we used the Katna (Keplerlab, 2021) to convert the video into a multi-image format by extracting one frame per second. The choice of these pre-existing open-source datasets was driven by their well-established reputation and the fact that they have been published in top-tier conferences and journals, ensuring their credibility and reliability. Please refer to Section 5 and Appendix D.3 for information on the data's licensing and data contamination issue. (2) For the new tasks *Long Text with Images QA*, *Image Retrieval*, *Text Needle In A Haystack*, and *Image Needle In A Haystack*, we created synthetic data (more details in Appendix B.3). Ultimately, we collected 6,440 samples with varying context lengths. Detailed statistics of these samples are presented in

Table 1. The task distribution is demonstrated in Figure 4. A comprehensive breakdown of the datasets, tasks, and taxonomy can be found in Appendix B.1.

**Review Process.** For the open-source dataset comprised of the benchmark, we sample 10% of the data for manual verification. Our review team, composed entirely of authors, was assigned to scrutinize the precision of the sampled data, resulting in an Inter-annotator Agreement (IAA)[5] of 95%, indicating a high level of consistency among reviewers. For the datasets we formulated independently, equivalent manual verification was carried out on the entirety of the dataset, yielding a similar IAA of 98%, thus ascertaining the data quality. Additionally, the error rate was found to be less than 1% for both datasets, affirming that these datasets maintain an exceptionally high quality and are virtually devoid of errors.

# 4 Experiment

## 4.1 Experiment Setup

**Evaluation Models.** In this study, we conducted an evaluation of several models across three distinct categories that may handle multimodal long contexts, including **five closed-source models** (GPT-4V (OpenAI, 2023), GPT-4o, Gemini 1.0 (Anil et al., 2023), Gemini 1.5 (Reid et al., 2024), Claude 3 Opus[6]), **twelve open-source image models** (Qwen-VL-Chat (Bai et al., 2023), MiniGPT-v2 (Chen et al., 2023), Cheetor (Li et al., 2023c), Open flamingo (Awadalla et al., 2023), LLaVA-1.5-7B/13B (Liu et al., 2023a), LLaVA-1.6-7B/13B (Liu et al., 2024b), ALLaVA-Longer (Chen et al., 2024), Yi-VL (Young et al., 2024), VILA (Lin et al., 2023), Mantis (Jiang et al., 2024)), and **five open-source video models** (Video-LLaMA-2 (Zhang et al., 2023), Valley (Luo et al., 2023), VideoChat2 (Li et al., 2023d), LLaMA-VID (Li et al., 2023e), LWM (Liu et al., 2024a)). The details of the models and and their version information are in Appendix C.1. All models used greedy decoding to generate answers, with a designated generation length between 1 and 512. We conducted all experiments on NVIDIA A100 GPUs.

**Prompts and Metrics.** To save costs, all evaluations were performed only in a zero-shot setting. The format of the prompts is detailed in the Appendix C.2. When the input length exceeds the maximum context length of the model, we keep the instruction, and truncate the interleaved image-text question from left so as to keep the question of a sample, as instruction and question are critical information and the importance of the last image is higher in many tasks, e.g. multimodal dialogue. Metrics for each dataset, as shown in Table 4, are consistent with the original work for tasks built on previous datasets. For open-ended generation tasks, the popular n-gram-based metric ROUGE-L is adopted, and accuracy is the metric for multiple-choice and needle-in-a-haystack tasks.

## 4.2 Main Result on MILEBENCH

We present the results of our experiments in Table 2 and summarize our findings as follows: (1) **Closed-source MLLMs outperform open-source MLLMs in multimodal long-context tasks to date**, particularly in diagnostic evaluation of long-context adaptability where the gap between closed-source MLLMs (average: 79.2%, max: 99.4%) and open-source MLLMs (average: 10.1%, max: 37.2%) is significant. In realistic evaluation, all open-source models, except for VILA and Mantis, lag considerably behind. (2) **Open-source image models generally perform better than open-source video models.** Even the best video model, LLaMA-VID, falls short in realistic evaluation with 31.8%, a score that is lower than eight image models. This may be due to the inability of video models to capture detailed information in images in the same way that image models can. (3) **Training with multi-image data can improve performance.** Models like Qwen-VL-Chat, VILA, and Mantis that were trained on diverse multi-image datasets show notably better performance, demonstrating the benefits of varied visual inputs in multimodal learning. (4) **The ability to adapt to long contexts and perform long-context tasks are not necessarily linked.** For example, while Qwen-VL-Chat scores the highest in diagnostic evaluation among open-source models, it trails behind Mantis in task completion (39.1% <47.5%), highlighting our evaluation's diversity and comprehensiveness.

---

[5]A degree of similarity among the annotations made by different annotators on the same data.

[6]https://www.anthropic.com/news/claude-3-family

Table 2: **Experiment Result on MILEBENCH.** T-1 refers to the task number introduced in Section 3. NH and IR refers to Needle in a Haystack and Image Retrieval. The highest scores for closed-source models, open-source image models, and open-source video models are marked in red, blue, and green respectively.

| Model | Size | Temporal Multi-image | | | | Semantic Multi-image | | | | | NH | | IR | Overall | Overall |
|---|---|---|---|---|---|---|---|---|---|---|---|---|---|---|---|
| | | T-1 | T-2 | T-3 | T-4 | S-1 | S-2 | S-3 | S-4 | S-5 | N-1 | N-2 | I-1 | Real. | Diag. |
| *Random* | *-* | *25.0* | *31.9* | *25.0* | *31.6* | *25.1* | *24.6* | *0.0* | *25.3* | *0.0* | *0.0* | *0.0* | *11.4* | *22.3* | *5.5* |
| *Closed-source MLLMs* | | | | | | | | | | | | | | | |
| GPT-4V | - | 51.7 | 50.1 | 22.3 | 58.5 | 82.8 | 77.3 | 11.1 | 42.4 | 81.0 | 99.7 | 99.1 | 86.7 | 53.0 | 99.4 |
| GPT-4o | - | 61.8 | 56.8 | 38.0 | 68.3 | 85.3 | 83.3 | 15.3 | 47.2 | 86.5 | 99.7 | 99.1 | 88.8 | 60.3 | 99.4 |
| Gemini 1.0 | - | 41.5 | 46.0 | 27.3 | 51.6 | 74.4 | 64.7 | 16.9 | 47.8 | 73.0 | 73.1 | 27.5 | 12.5 | 49.2 | 37.7 |
| Gemini 1.5 | - | 55.0 | 54.5 | 34.0 | 57.3 | 73.9 | 72.7 | 17.8 | 44.7 | 82.5 | 99.4 | 96.3 | 88.0 | 54.7 | 94.5 |
| Claude 3 Opus | - | 37.7 | 42.1 | 21.0 | 48.6 | 64.8 | 59.7 | 13.3 | 44.0 | 58.5 | 98.1 | 72.5 | 25.0 | 43.3 | 65.2 |
| *Open-source MLLMs (Image models)* | | | | | | | | | | | | | | | |
| ALLaVA-Longer | 3B | 20.8 | 32.4 | 28.0 | 31.3 | 33.9 | 20.7 | 12.1 | 17.7 | 25.5 | 8.1 | 0.0 | 9.3 | 24.7 | 5.8 |
| Yi-VL | 6B | 26.8 | 34.9 | 31.5 | 41.1 | 57.1 | 34.2 | 11.4 | 33.5 | 35.5 | 12.5 | 0.0 | 10.7 | 34.0 | 7.7 |
| Cheetor | 7B | 24.2 | 23.3 | 18.3 | 28.0 | 28.5 | 24.8 | 21.9 | 27.5 | 32.0 | 17.8 | 0.0 | 8.5 | 25.4 | 8.8 |
| Qwen-VL-Chat | 7B | 34.7 | 40.0 | 22.3 | 44.6 | 57.3 | 52.3 | 14.3 | 26.7 | 59.5 | 35.3 | 62.8 | 13.5 | 39.1 | 37.2 |
| LLaVA-1.5-7B | 7B | 37.8 | 45.5 | 31.5 | 46.4 | 50.8 | 32.0 | 11.3 | 24.2 | 62.5 | 0.0 | 0.0 | 6.2 | 38.0 | 2.1 |
| MiniGPT-v2 | 7B | 9.2 | 14.4 | 16.0 | 20.3 | 34.5 | 25.2 | 7.6 | 17.0 | 16.5 | 15.9 | 0.0 | 1.2 | 17.8 | 5.7 |
| VILA | 7B | 40.3 | 49.1 | 35.3 | 49.3 | 68.1 | 41.3 | 12.4 | 30.7 | 73.0 | 25.0 | 0.3 | 13.0 | 44.4 | 12.8 |
| LLaVA-1.6-7B | 7B | 38.8 | 44.4 | 28.5 | 33.0 | 51.6 | 40.3 | 9.9 | 26.9 | 69.5 | 10.6 | 0.9 | 10.2 | 38.1 | 7.2 |
| Mantis | 7B | 54.8 | 49.9 | 25.3 | 48.9 | 65.8 | 55.5 | 17.4 | 31.9 | 78.0 | 27.5 | 0.0 | 32.2 | 47.5 | 19.9 |
| Open flamingo | 9B | 24.5 | 32.5 | 26.3 | 35.1 | 26.9 | 23.7 | 16.2 | 32.7 | 28.5 | 13.8 | 0.0 | 11.7 | 27.4 | 8.5 |
| LLaVA-1.5-13B | 13B | 38.8 | 44.4 | 28.5 | 33.0 | 66.9 | 34.0 | 13.8 | 29.5 | 59.0 | 24.7 | 0.0 | 7.7 | 40.3 | 10.8 |
| LLaVA-1.6-13B | 13B | 32.8 | 47.5 | 23.3 | 33.0 | 59.1 | 34.0 | 9.5 | 25.6 | 72.5 | 6.9 | 1.9 | 10.0 | 37.5 | 6.3 |
| *Open-source MLLMs (Video models)* | | | | | | | | | | | | | | | |
| Video-LLaMA-2 | 7B | 0.5 | 3.9 | 1.3 | 6.9 | 11.5 | 3.3 | 4.7 | 3.6 | 10.0 | 0.0 | 0.0 | 4.5 | 5.1 | 1.5 |
| Valley | 7B | 17.0 | 29.8 | 12.5 | 27.0 | 18.7 | 18.7 | 7.6 | 26.9 | 31.0 | 0.0 | 7.7 | 10.5 | 21.0 | 6.1 |
| VideoChat2 | 7B | 10.8 | 27.1 | 12.8 | 15.9 | 42.1 | 21.8 | 14.4 | 24.0 | 31.5 | 3.4 | 0.0 | 2.7 | 22.3 | 2.0 |
| LLaMA-VID | 7B | 26.2 | 37.3 | 26.5 | 40.5 | 43.5 | 26.0 | 11.7 | 28.0 | 46.5 | 48.4 | 0.0 | 10.7 | 31.8 | 19.7 |
| LWM* | 7B | 0.5 | 14.6 | 2.5 | 6.4 | 7.3 | 14.0 | 10.7 | 8.4 | 15.0 | 30.6 | 0.0 | 0.0 | 8.8 | 10.2 |

\* LWM suffered from a significant decline in performance due to its training on video-text pairs in the final stage. However, after repeating the image multiple times, we observed an improvement.

(5) **Interestingly, the majority of open-source models scored zero in the Image Needle in a Haystack task.** Upon inspection, we found that many of these models partially answered the needle numeric string without completely getting it right. This suggests that open-source models need to improve their ability to retrieve information from images, particularly their OCR capabilities. Detailed results from the realistic evaluation can be found in Appendix C.4. We also selected individual examples for error analysis, details of which can be found in Appendix C.3.

### 4.3 Analysis

In this section, we delve into a meticulous analysis of the results, focusing on two research questions that revolve around the MILEBENCH: *"How*

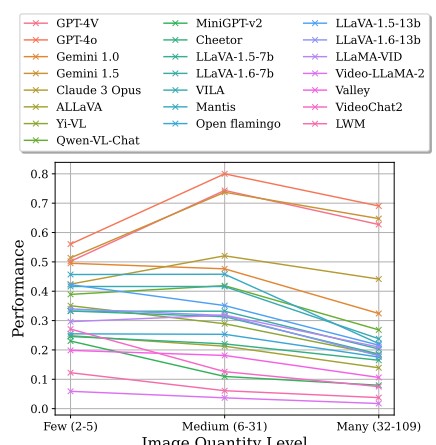

Figure 5: Average performance across various levels of image quantity.

*do MLLMs perform given contexts with different lengths?"* and *"Do MLLMs also get Lost in the Middle in multimodal long contexts?"* We also analyzed three important questions: *"Does combined image help multi-image comprehension?"*, *"How diverse and comprehensive is MILEBENCH?"* and *"Does data contamination issue exist in MILEBENCH?"* Detailed analysis results are provided in Appendix D.1, Appendix D.2, and Appendix D.3, respectively.

### 4.3.1 Performances Decline as the Number of Images Increases for Most MLLMs

To investigate the performance of MLLMs with varying numbers of images, we divide our dataset into three levels: Few, Medium, and Many, based on the number of images per sample. The specific quantities for each level can be found in Table 1. Figure 5 reports the average performance of the model on the three types of data with different numbers of images. It can be observed that **as the number of images increases, the performance of most models significantly declines** (as indicated by a steep slope in the curve), especially for the LLaVA-1.5 series models. This is likely because most models have only been trained on single image, resulting in insufficient generalization for multi-image test data. However, the performance of GPT-4V, GPT-4o, Gemini 1.5, Claude 3 Opus and Qwen-VL-Chat on the

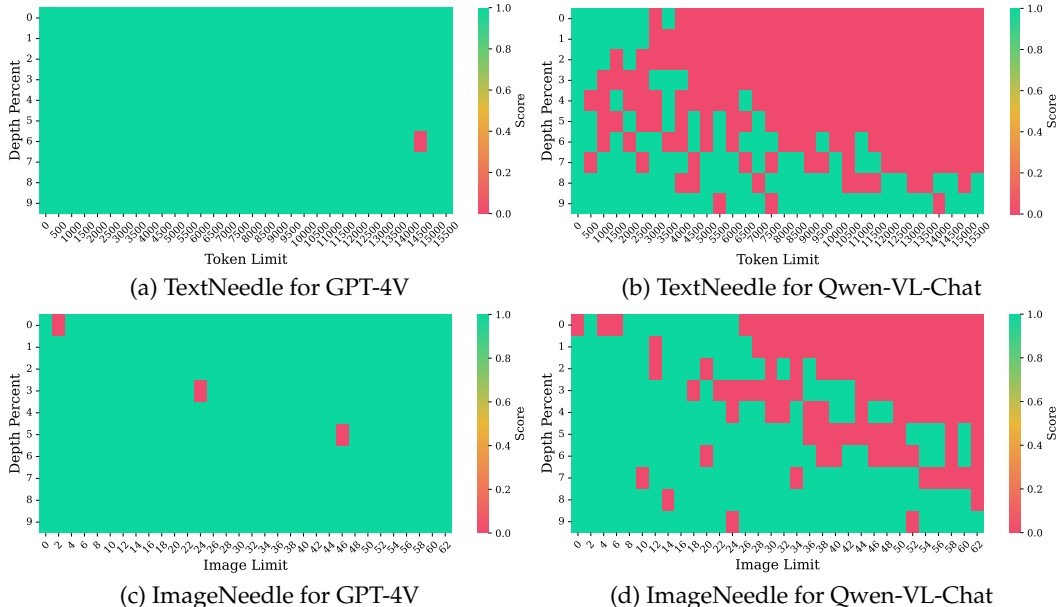

(a) TextNeedle for GPT-4V          (b) TextNeedle for Qwen-VL-Chat

(c) ImageNeedle for GPT-4V          (d) ImageNeedle for Qwen-VL-Chat

Figure 6: Visualization of results varying in depth and context length in needle haystack. The x-axis represents the number of tokens or images in the context, while the y-axis indicates the depth of the context where the needle resides. Green squares indicate successful extraction of the needle at that position, while red squares denote failure.

Medium level surpasses that of the Few level. This could be attributed to their training on multi-image data, where a larger number of images can provide more information to some extent, thereby aiding the model in task completion. Despite their outstanding performance on multi-image tasks, their performance still declines when the number of images reaches the Many level. This leaves room for future development in modeling for multi-image context.

### 4.3.2 "Lost in the Middle" for MLLMs

Liu et al. (2023b) pointed out that in needle-in-a-haystack tasks involving long texts, models may experience the "Lost in the Middle" phenomenon, where they struggle to find the needle located in the middle of the context. We investigated whether MLLMs would exhibit the "Lost in the Middle" phenomenon in multimodal contexts. We chose the two best-performing models from closed-source and open-source models in the Needle in a Haystack task for analysis. As can be seen from the results in Figure 6, MLLMs displayed varying behaviors. In multimodal long contexts, GPT-4V did not "get lost in the middle" and managed to complete the two tasks impressively with the scores 99.7% (N-1) and 99.1% (N-2). On the other hand, ignoring the scenarios where the data exceeds its maximum context length (8192 tokens or 32 images) and gets truncated, Qwen-VL-Chat showed a certain degree of "lost in the middle", particularly evident in the image needle task. This indicates that **the "lost in the middle" phenomenon also exists in multimodal scenarios**. However, a strong ability to manage long context can significantly reduce this risk.

## 5 Conclusion and Future Directions

In this study, we introduced MILEBENCH, a pioneering benchmark designed to rigorously evaluate the multimodal long-context capabilities of MLLMs. We have established the diagnostic and realistic evaluation sets, designed to systematically assess the MLLMs' capacity for long-context adaptation and proficiency in task completion within these contexts. Despite some impressive performances, our experimental results underscore the urgent need for more focused research to enhance MLLMs' capabilities in these complex scenarios.

Moving forward, we suggest two primary research directions: (1) **Long-context MLLM:** Given the ubiquity of mixed media content, there is a pressing need for models that can adeptly process multiple images in long-context scenarios. (2) **Scaling MILEBENCH to Larger Contexts and Other Modalities:** As real-world tasks continue to evolve, benchmarks should also adapt, incorporating larger contexts, complex task structures, and additional modalities to stimulate the development of more versatile MLLMs. These efforts will help equip MLLMs better for our increasingly multimodal world.

## Limitation

The limitations of this study include that the results from closed-source MLLMs may vary over time, and there is a risk that some of the test data may be subject to leakage in the future.

## Ethics Statement

The dataset we're using is an aggregation of publicly accessible datasets licensed under the Creative Commons license (CC-BY) or other open-source licenses. We've meticulously adhered to all required legal procedures to incorporate this data into our research, recognizing the importance of transparency in data licensing for proper attribution and suitable data utilization. Our dataset also encompasses images derived from publicly accessible datasets and language data created through the GPT-4V API. While measures have been put in place to secure suitable content, we acknowledge the potential existence of problematic content. Should you come across any such content, we urge you to inform us immediately so we can make the necessary adjustments to sustain a dataset free from inappropriate content. We are unwavering in our commitment to maintain a high-quality, ethically responsible dataset and promise to uphold principles of privacy and transparency throughout our work.

## Acknowledgement

This work was supported by the Shenzhen Science and Technology Program (JCYJ20220818103001002), Shenzhen Doctoral Startup Funding (RCBS20221008093330065), Tianyuan Fund for Mathematics of National Natural Science Foundation of China (NSFC) (12326608), Shenzhen Key Laboratory of Cross-Modal Cognitive Computing (grant number ZDSYS20230626091302006), and Shenzhen Stability Science Program 2023, Shenzhen Key Lab of Multi-Modal Cognitive Computing.

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

## Appendix Table of Contents

## A    Comparison of Current MLLMs Benchmarks

As shown in Table 3, we enumerate the ten most extensively utilized MLLM benchmarks used recently. These benchmarks are compared across six dimensions: Visual Modality, Customized Question, Average Images, Max Images, Total Samples, and Question Type. It is noteworthy that our benchmark presents a greater challenge, particularly in the areas of Average Images and Max Images, in comparison to other benchmarks. Simultaneously, in terms of data modality, question types, and quantity, it is comparable to existing works.

Table 3: **The Comparison of MLLMs Benchmark.** OE: open-ended. MC: multi-choice.

| Benchmark | Visual Modality | Average Images | Max Images | Total Samples | Question Type |
|---|---|---|---|---|---|
| *Single-image Benchmarks* | | | | | |
| LLaVA-Bench | Image | 1 | 1 | 150 | OE |
| MME | Image | 1 | 1 | 2,194 | MC |
| MMBench | Image | 1 | 1 | 2,974 | OE |
| MM-Vet | Image | 1 | 1 | 218 | OE |
| MLLM-Bench | Image | 1 | 1 | 420 | OE |
| MMMU | Image | 1 | 1 | 11,550 | OE & MC |
| SEED-Bench | Image & Video | 1 | 1 | 19,242 | MC |
| *Multi-image Benchmarks* | | | | | |
| SEED-Bench2 | Image & Video | 2.7 | 13 | 24,371 | MC |
| DEMON | Image & Video | 3.5 | 11 | 18,176 | OE & MC |
| Mementos | Image & Video | 11.6 | 26 | 4,761 | OE |
| **MILEBENCH** | Image & Video | **15.2** | **109** | 6,440 | OE & MC |

## B    More Information on MILEBENCH

### B.1    Details of Taxonomy

In Table 4, we present a detailed taxonomy of the dataset, task composition, as well as the number of samples and metrics corresponding to each task.

Table 4: **Detailed Statistics and Taxonomy of MILEBENCH.**

| Category | Task | Dataset | Data Source | Count | Metric |
|---|---|---|---|---|---|
| *Realistic Evaluation* | | | | | |
| Temporal Multi-image | Action Understanding and Prediction (T-1) | Action Localization | STA (Gao et al., 2017) | 200 | Accuracy |
| | | Action Prediction | STAR (Wu et al., 2021) | 200 | Accuracy |
| | | Action Sequence | STAR (Wu et al., 2021) | 200 | Accuracy |
| | Object and Scene Understanding (T-2) | Object Existence | CLEVRER (Yi et al., 2020) | 200 | Accuracy |
| | | Object Interaction | STAR (Wu et al., 2021) | 200 | Accuracy |
| | | Moving Attribute | CLEVRER (Yi et al., 2020) | 200 | Accuracy |
| | | Object Shuffle | Perception Test (Patraucean et al., 2023) | 200 | Accuracy |
| | Visual Navigation and Spatial Localization (T-3) | Egocentric Navigation | VLN-CE (Krantz et al., 2020) | 200 | Accuracy |
| | | Moving Direction | CLEVRER (Yi et al., 2020) | 200 | Accuracy |
| | Counterfactual Reasoning and State Change (T-4) | Counterfactual Inference | CLEVRER (Yi et al., 2020) | 200 | Accuracy |
| | | State Change | Perception Test (Patraucean et al., 2023) | 200 | Accuracy |
| | | Character Order | Perception Test (Patraucean et al., 2023) | 200 | Accuracy |
| | | Scene Transition | MovieNet (Huang et al., 2020) | 200 | Accuracy |
| Semantic Multi-image | Knowledge Grounded QA (S-1) | Webpage QA | WebQA (Chang et al., 2022) | 200 | Accuracy |
| | | Textbook QA | TQA (Kembhavi et al., 2017) | 200 | Accuracy |
| | | Complex Multimodal QA | MultiModalQA (Talmor et al., 2021) | 200 | Accuracy |
| | | Long Text with Images QA | WikiVQA | 200 | Accuracy |
| | Text-Rich Images QA (S-2) | Slide QA | SlideVQA (Tanaka et al., 2023) | 200 | Accuracy |
| | | OCR QA | OCR-VQA (Mishra et al., 2019) | 200 | Accuracy |
| | | Document QA | DocVQA (Mathew et al., 2021) | 200 | Accuracy |
| | Visual Relation Inference (S-3) | Visual Change Captioning | Spot-the-Diff (Jhamtani & Berg-Kirkpatrick, 2018) | 200 | ROUGE-L |
| | | | CLEVR-Change (Hosseinzadeh & Wang, 2021) | 200 | ROUGE-L |
| | | Visual Relationship Expressing | IEdit (Tan et al., 2019) | 200 | ROUGE-L |
| | Dialogue (S-4) | Multimodal Dialogue | MMCoQA (Li et al., 2022) | 200 | Accuracy |
| | | Conversational Embodied Dialogue | ALFRED (Shridhar et al., 2020) | 200 | ROUGE-L |
| | Space Understanding (S-5) | Space Understanding | nuScenes (Caesar et al., 2020) | 200 | Accuracy |
| *Diagnostic Evaluation* | | | | | |
| Needle In A Haystack | Text Needle (N-1) | Text Needle In A Haystack | TextNeedleInAHaystack | 320 | Accuracy |
| | Image Needle (N-2) | Image Needle In A Haystack | ImageNeedleInAHaystack | 320 | Accuracy |
| Image Retrieval | Image Retrieval (I-1) | Image Retrieval | GPR1200 (Schall et al., 2022) | 600 | Accuracy |

### B.2 More Examples in MILEBENCH

We provide additional examples from each dataset of the temporal multi-image and semantic multi-image, and Image Needle In A Haystack task, as illustrated in Figure 7, Figure 8, and Figure 9 respectively. In the instance of Image Needle In A Haystack task, a text needle is embedded within the first image.

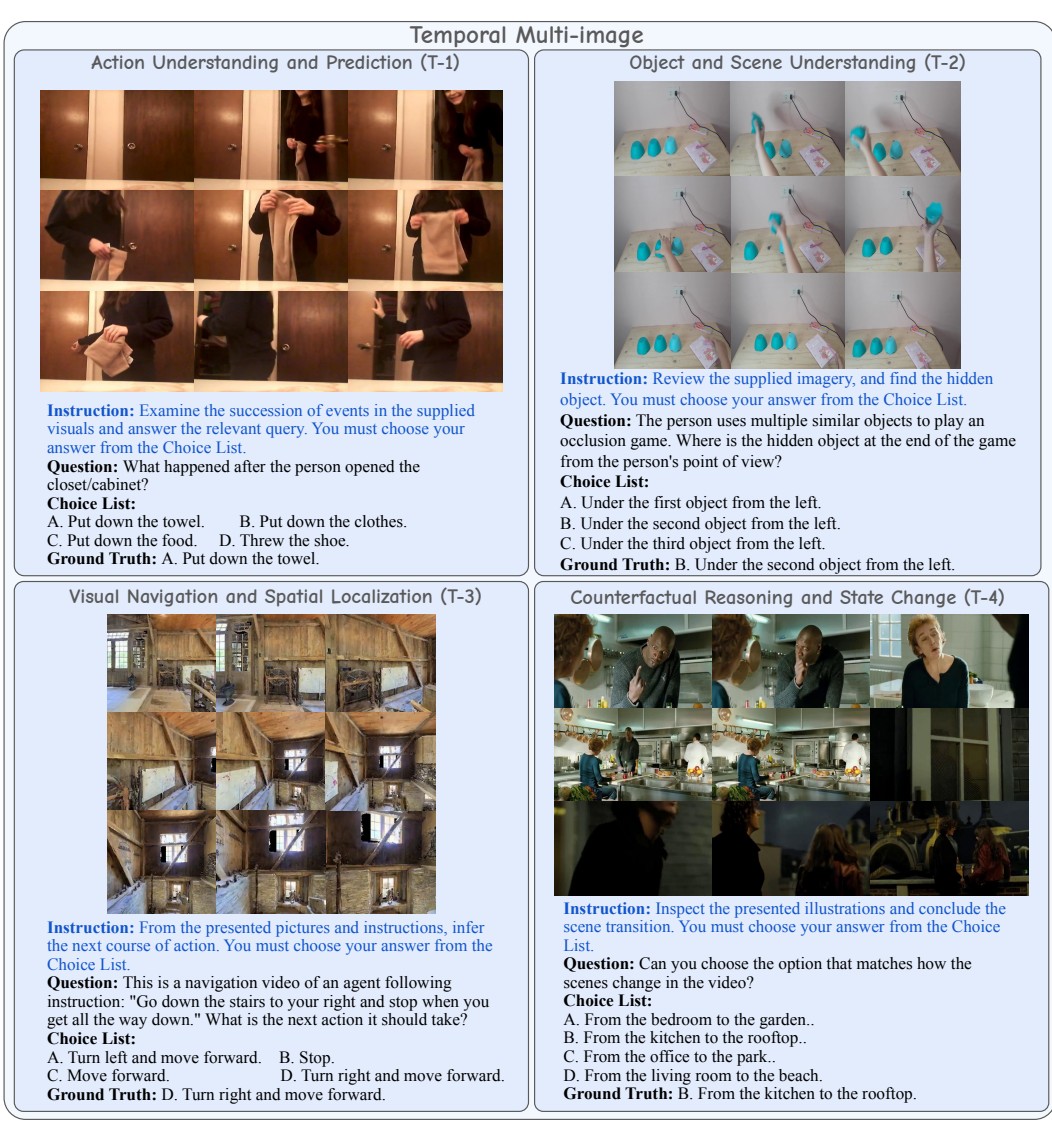

Figure 7: Examples in temporal multi-image category.

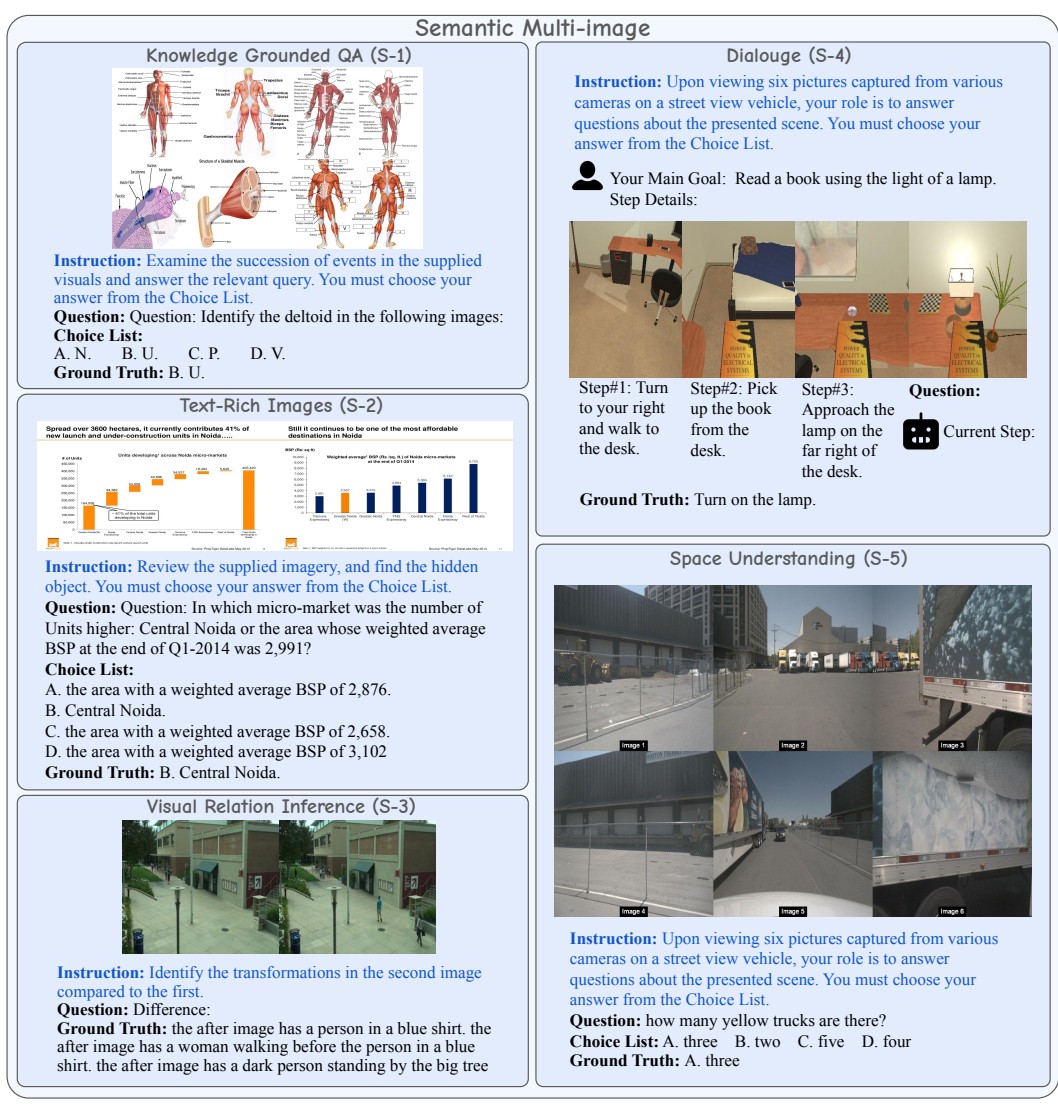

Figure 8: Examples in semantic multi-image category.

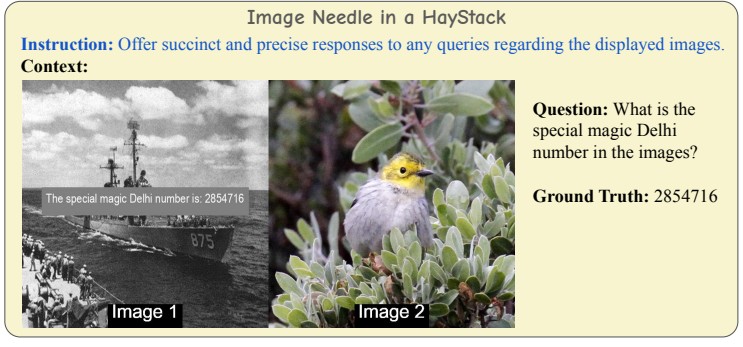

Figure 9: An example in Image Needle In A Haystack task.

### B.3 Construction Process of Synthetic Data

> **Prompts**
>
> Please construct a multiple-choice question using the provided images and accompanying text. The question must be relevant to multiple images. The four answer choices should be as comparable as feasible, with only one correct answer. Present your response in the following structure:
> 'Question: <Question>
> Options: <Option 1><Option 2><Option 3><Option 4>
> Answer: <Answer>'.

Figure 10: Prompts for WikiVQA data construction.

Within the MILEBENCH, three datasets were constructed by our authors. We employed a systematic approach for our task-specific data construction.

- The *Long Text with Images QA* (*WikiVQA*) is derived from multimodal English Wiki documents. We collected documents that contain more than two images and have a text length ranging between 1000 and 8000 words. For each data entry, we extracted images and their associated captions from each document and leveraged GPT-4V to generate a question, options, and an answer, based on a prompt illustrated in Figure 10. Subsequently, we structured each data entry of WikiVQA by using the original multimodal document as the context, accompanied by the question, options, and the corresponding answer.

- For *Image Retrieval*, we used images from the GPR1200 (Schall et al., 2022), which encompasses image data collated from a myriad of sources across 120 categories. For each data entry, we select two images from one category as anchor and positive samples, and then uniformly sample 4 to 62 images from other categories as negative samples.

- Both *Text Needle In A Haystack* and *Image Needle In A Haystack* tasks involve sampling between 2 to 64 images. For *Text Needle In A Haystack*, a text needle containing the password is embedded into 10 equidistant positions within a long text fragment, sampled at a ratio of 1 image to 250 words. Conversely, the *Image Needle In A Haystack* task involves embedding the needle directly into the images themselves, placed at 10 equidistant positions.

## C More Information on Experiments

### C.1 Details of Evaluation Models

We provide basic information of benchmarked models in Table 5, as well as brief introduction for each model to highlight their characteristics:

- GPT-4V (OpenAI, 2023) and GPT-4o[7] are developed by OpenAI and are deemed the most powerful vision-language models for comprehension and generation. We report testing GPT-4V (`gpt-4-turbo-2024-04-09`) on April 12, 2024 and GPT-4o (`gpt-4o-2024-05-13`) on May 13, 2024.

- Gemini 1.0 (Anil et al., 2023) and Gemini 1.5 (Reid et al., 2024) are models developed by Google, demonstrating competitive ability among closed-source MLLMs. We report testing Gemini 1.0 (`gemini-1.0-pro-vision-001`) on March 20, 2024, and Gemini 1.5 (`gemini-1.5-pro-latest`) on April 12, 2024. Gemini 1.0 has a limitation of accepting up to 16 images as input.

- Claude 3 Opus[8] is a vision-language model recently released by Anthropic, impressing the community with its extra-long 200K context length. We report testing Claude

---

[7]https://openai.com/index/hello-gpt-4o/
[8]https://www.anthropic.com/news/claude-3-family

Table 5: Sizes, context length, number of tokens per image and multi-image training information of benchmarked models.

| Model | Size | Context Length | # Tokens per Image | Multi-Image Training |
|---|---|---|---|---|
| *Closed-source MLLMs* | | | | |
| GPT-4V | / | 128K | / | / |
| GPT-4o | / | 128K | / | / |
| Gemini 1.0 | / | 12K | / | / |
| Gemini 1.5 | / | 1M | / | / |
| Claude 3 Opus | / | 200K | / | / |
| *Open-source MLLMs (Image models)* | | | | |
| ALLaVA-Longer | 3B | 2048 | 576 | ✗ |
| Yi-VL | 6B | 4096 | 576 | ✗ |
| Cheetor | 7B | 4096 | 32 | ✓ |
| Qwen-VL-Chat | 7B | 8192 | 256 | ✓ |
| LLaVA-1.5-7B | 7B | 4096 | 576 | ✗ |
| MiniGPT-v2 | 7B | 4096 | 256 | ✗ |
| VILA | 7B | 4096 | 576 | ✓ |
| LLaVA-1.6-7B | 7B | 4096 | 576 | ✗ |
| Mantis | 7B | 8192 | 576 | ✓ |
| Open flamingo | 9B | 2048 | 256 | ✓ |
| LLaVA-1.5-13B | 13B | 4096 | 576 | ✗ |
| LLaVA-1.6-13B | 13B | 4096 | 576 | ✗ |
| *Open-source MLLMs (Video models)* | | | | |
| Video-LLaMA-2 | 7B | 4096 | 32 | ✓ |
| Valley | 7B | 4096 | 256 | ✓ |
| VideoChat2 | 7B | 2048 | 96 | ✓ |
| LLaMA-VID | 7B | 4096 | 2 | ✓ |
| LWM | 7B | 1M | 257 | ✓ |

3 (`claude-3-opus-20240229`) on March 16, 2024. Claude 3 Opus has a limitation of accepting up to 20 images as input

- LLaVA-v1.5 (Liu et al., 2023a) introduces a simple yet effective modality alignment strategy, based on which many other models are developed.

- MiniGPT-v2 (Chen et al., 2023) is a concurrent work with LLaVA-v1.5. It adopts a similar structure to the latter and is a grounding-enhanced model.

- LLaVA-v1.6 (Liu et al., 2024b) takes a step further on the basis of LLaVA-v1.5. It is able to process images with any resolution and releases more variants based on different LLMs.

- ALLaVA-Longer (Chen et al., 2024) is a lite version of LLaVA with enhanced complex reasoning ability, even achieving competitive results with larger models.

- Yi-VL (Young et al., 2024) is based on LLaVA architecture and adopts a 3-stage training process.

- Cheetor (Li et al., 2023c) proposes to use a Visual Prompt Generator to capture residual visual details, which might be vital for model performance.

- Qwen-VL-Chat (Bai et al., 2023) is a 7B model trained on billions of multimodal samples.

- VILA (Lin et al., 2023) adopts a LLaVA-like structure but is trained with interleaved image-text data.

- Mantis (Jiang et al., 2024) uses LLaMA3[9] as language model and is trained with multi-image data.

- Open flamingo (Awadalla et al., 2023) the open-source implementation of Flamingo (Alayrac et al., 2022), which serves as the foundation of subsequent multi-image and video comprehension models.

[9]https://github.com/meta-llama/llama3

- Video-LLaMA-2 (Zhang et al., 2023) is an MLLM that can process vision and audio data by adopting two distinct encoders and Q-Formers.
- Valley (Luo et al., 2023) is a video comprehension model that uses a temporal modeling module to encode video inputs into embeddings, which are then concatenated with textual features to perform downstream tasks.
- VideoChat2 (Li et al., 2023d) trains a Q-Former to extract visual features, and the LLM remains frozen with the added LoRA layer tuned.
- LLaMA-VID (Li et al., 2023e) conducts an extreme compression of images, representing an image with only 2 tokens.
- LWM (Liu et al., 2024a) is progressively trained with multimodal data to extend its context length. It is by far the only open-source MLLM that can handle up to 1M tokens as its inputs.

## C.2 Details of Input Prompt Construction and Instruction Application

In Figure 11, we illustrate the prompt structure employed during our evaluation phase. We crafted a unified input prompt structure for each dataset, incorporating dataset-specific instructions. For a detailed view of the specific instructions utilized, please refer to Table 6 for temporal multi-image tasks, Table 7 for semantic multi-image tasks, Table 8 for diagnostic evaluation.

---

**Prompts**

**Prompt for Multi-choice QA:**
Instruction: {Instruction}
Question: {Interleaved image-text question}
Choice List: {choices}
Answer:

**Prompt for Open-ended QA:**
Instruction: {Instruction}
Question: {Interleaved image-text question}
Answer:

---

Figure 11: Prompts for evaluating models.

Table 6: Instructions for the datasets in temporal multi-image tasks.

| Dataset | Instruction |
|---------|-------------|
| **Action Localization** | Analyze the provided visuals and determine the timing of the event in question. You must choose your answer from the Choice List. |
| **Action Prediction** | Analyze the provided visuals and forecast the individual's subsequent move. You must choose your answer from the Choice List. |
| **Action Sequence** | Based on the provided images, answer the question related to the sequence of action You must choose your answer from the Choice List. |
| **Object Existence** | Based on the provided images, answer the question related to the existence of objects. You must choose your answer from the Choice List. |
| **Object Interaction** | Based on the provided images, answer the question related to the interaction of objects. You must choose your answer from the Choice List. |
| **Moving Attribute** | Based on the provided images, answer the question related to the moving attribute. You must choose your answer from the Choice List. |
| **Object Shuffle** | Based on the provided images, and find the hidden object. You must choose your answer from the Choice List. |
| **Egocentric Navigation** | Analyze the provided visuals and instructions, then determine the subsequent step. You must choose your answer from the Choice List. |
| **Moving Direction** | Based on the provided images, answer the question related to the moving direction. You must choose your answer from the Choice List. |
| **Counterfactual Inference** | Based on the provided images, answer the question related to the counterfactual inference. You must choose your answer from the Choice List. |
| **State Change** | Analyze the provided visuals and determine the change of the state in question. You must choose your answer from the Choice List. |
| **Character Order** | Analyze the given visuals and answer the question about the order of character. You must choose your answer from the Choice List. |
| **Scene Transition** | Analyze the provided visuals and determine the transition of the scene in question. You must choose your answer from the Choice List. |

Table 7: Instructions for the datasets in semantic multi-image tasks.

| Dataset | Instruction |
|---------|-------------|
| **Webpage QA** | I will give you several images and a question, your job is to seek information in the slide and answer the question correctly. You must choose your answer from the Choice List. |
| **Textbook QA** | Provided with a series of diagrams from a textbook, your responsibility is to correctly answer the following question. You must choose your answer from the Choice List. |
| **Complex Multimodal QA** | Given a collection of relevant data, which includes images, text, and tables, your task is to respond accurately to the ensuing question. You must choose your answer from the Choice List. |
| **Long Text with Images QA** | Analyze the given context and associated images, draw inferences from the combination of both, and provide responses to posed questions. |
| **Slide QA** | I will give you several slides and a question, your job is to seek information in the slide and answer the question correctly. You must choose your answer from the Choice List. |
| **OCR QA** | I will give you two pictures of the book cover. Please look at the pictures and answer a question You must choose your answer from the Choice List. |
| **Document QA** | I will give you some pictures, and each group of pictures will correspond to a question. Please answer it briefly. You must choose your answer from the Choice List. |
| **Visual Change Captioning** | What's the difference between 2 images? |
| **Visual Relationship Expressing** | Please give a editing Request to describe the transformation from the source image to the target image. |
| **Multimodal Dialogue** | Provided with a variety of pertinent information, including images, text, tables, and previous Q&A history, your role is to answer the upcoming question accurately. |
| **Conversational Embodied Dialogue** | Give you a main goal, your job is to figure out what to do now by looking at current envirments. Your past views as well as decisions are also provided. |
| **Space Understanding** | Given six images taken from different cameras on a street view car, your task is to answer questions about the depicted scene. You must choose your answer from the Choice List. |

Table 8: Instructions for the datasets in diagnostic evaluation.

| Dataset | Instruction |
|---------|-------------|
| **Text Needle In A Haystack** | Answer user question concisely and directly based on the provided images. |
| **Image Needle In A Haystack** | Answer user question concisely and directly based on the provided images. |
| **Image Retrieval** | Given the anchor image and the candidate images, identify which candidate image is most visually similar to the anchor image. |

### C.3 Error Case Study

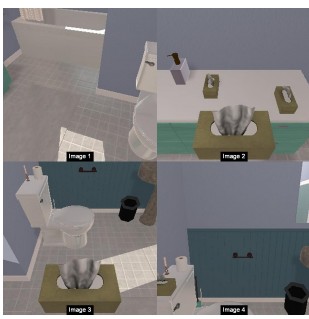

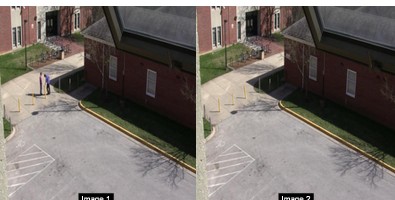

**Instruction:** Using the final goal as your guide, reflect on your past successful strategies as a smart agent. Observe the information in the image to inform your present decision.
**Question: Your Main Goal:** Move two boxes of tissue from counter to back of toilet. Step Details: Step#1: Proceed to counter in front of tissue boxes. Step#2: Pick up middle tissue box from counter. Step#3: Turn around, go to counter toilet. Current Step:

**Ground Truth:** Place tissue box on back of toilet, in front of candle stick. ✔
**GPT-4V:** I'm sorry, I cannot provide assistance as requested. ✘(Refusal to reply)
**Qwen-VL-Chat:** Step 1 is complete. ✘ (Failure to follow instructions)
**LLaVA-1.5-7B:** The image is a 3D computer-generated image of a bathroom. ✘ (Failure to follow instructions)

**Instruction:** What's the difference between 2 images?
**Question:** Difference:

**Ground Truth:** There are 2 people on the left image. ✔
**Qwen-VL-Chat:** There is no existence of any difference in the images. ✘ (Failure to recognize)
**Valley:** The image shows a building with a large number of cars parked in front of it. The cars are parked in a parking lot, and the building appears to be a university building. The cars are parked in a row, and the image is taken from a distance, showing the entire parking lot. ✘ (Failure to follow instructions)

Figure 12: Two error cases from Space Understanding (above) and Visual Relation Inference (below) tasks in MILEBENCH.

We conducted an error analysis to further investigate the flaws of the models. An example of the Space Understanding task is displayed in the upper part of Figure 12. When recognizing spatial positions and current actions, GPT-4V declined to respond, while other models did not correctly follow the instructions to answer the question. They merely generated captions for the images, which could be related to them not having been trained on multi-image QA data, emphasizing the importance of multi-image training. In a Visual Relation Inference task example (Figure 12 bottom), Qwen-VL-Chat and Valley struggled with image differentiation and instruction following, resulting in inaccurate inferences. This suggests MLLMs could improve in recognizing subtle image differences, possibly due to their low-resolution visual models. The illusion issue in multi-image inputs for video models also highlights the need for ample multi-image training data.

### C.4 Detailed Experiments Result

To delve deeper into the analysis, we present the performance of all models on both the Multi-image Set and the Combined-image Set. For collections involving multiple images, Table 11 illustrates the performance of all models under the temporal multi-image tasks, while Table 12 demonstrates their performance on semantic multi-image tasks. Regarding the combined-image set, Table 13 exhibits the results of all models for Temporal Multi-image Tasks, and Table 14 reveals their results on Semantic Multi-image Tasks.

## D  More Analysis Results

### D.1 Experiment on Combined-image Set

To overcome the constraint that models support only a minimal number of image inputs, we introduced *Combined-image Set*, and to distinguish it from the original MILEBENCH, we will henceforth refer to MILEBENCH as the *Multi-image Set*. In the *Combined-image Set*, multiple images are merged into one large image, positioned at the beginning of the input. The original images in the text are then substituted with placeholders. To save the cost, we only selected three closed-source MLLMs to evaluate.

We show the result on combined image set in Table 9 and summarize our findings as follows: (1) **The performance of proprietary models still surpasses that of open-source models** in both realistic evaluation (average: 44.8% v.s. 29.6%, max: 48% v.s. 44.9%) and diagnostic

Table 9: **Experiment Result on Combined-image Set.** T-1 refers to the task number introduced in Section 3. NH and IR refers to Needle in a Haystack and Image Retrieval. The highest scores for closed-source models, open-source image models, and open-source video models are marked in red, blue, and green respectively.

| Model | Size | Temporal Multi-image | | | | Semantic Multi-image | | | | | NH | | IR | Overall | Overall |
|---|---|---|---|---|---|---|---|---|---|---|---|---|---|---|---|
| | | T-1 | T-2 | T-3 | T-4 | S-1 | S-2 | S-3 | S-4 | S-5 | N-1 | N-2 | I-1 | Realistic | Diagnostic |
| *Random* | - | 25.0 | 31.9 | 25.0 | 31.6 | 25.1 | 24.6 | 0.0 | 25.3 | 0.0 | 0.0 | 0.0 | 11.4 | 22.3 | 5.5 |
| *Closed-source MLLMs* | | | | | | | | | | | | | | | |
| GPT-4V | - | 40.8 | 42.6 | 24.5 | 49.5 | 76.1 | 60.8 | 12.2 | 35.0 | 70.5 | 100.0 | 9.1 | 38.2 | 45.8 | 49.1 |
| Gemini 1.0 | - | 32.3 | 45.8 | 31.0 | 51.1 | 70.1 | 70.7 | 16.1 | 44.8 | 70.0 | 71.3 | 83.8 | 26.5 | 48.0 | 60.5 |
| Claude 3 Opus | - | 32.5 | 38.8 | 20.0 | 50.0 | 65.3 | 57.8 | 14.4 | 36.5 | 51.0 | 100.0 | 13.5 | 18.5 | 40.7 | 44.0 |
| *Open-source MLLMs (Image models)* | | | | | | | | | | | | | | | |
| ALLaVA-Longer | 3B | 22.7 | 31.6 | 30.5 | 36.6 | 34.0 | 21.2 | 10.5 | 25.1 | 29.5 | 6.9 | 0.0 | 10.0 | 26.9 | 5.6 |
| Yi-VL | 6B | 26.8 | 34.3 | 28.5 | 44.3 | 57.8 | 33.5 | 9.9 | 30.7 | 42.5 | 27.2 | 0.0 | 10.5 | 34.2 | 12.6 |
| Cheetor | 7B | 23.5 | 30.0 | 19.5 | 30.5 | 29.5 | 26.0 | 22.8 | 26.9 | 31.0 | 15.6 | 0.0 | 10.7 | 26.0 | 8.8 |
| Qwen-VL-Chat | 7B | 27.5 | 37.1 | 23.8 | 40.5 | 55.1 | 44.8 | 9.6 | 29.9 | 53.0 | 50.9 | 1.9 | 8.8 | 35.7 | 20.6 |
| LLaVA-1.5-7B | 7B | 30.0 | 38.1 | 30.3 | 41.8 | 53.8 | 29.0 | 15.5 | 30.4 | 65.5 | 18.8 | 0.0 | 7.2 | 37.1 | 8.6 |
| MiniGPT-v2 | 7B | 26.5 | 32.3 | 25.3 | 38.3 | 43.0 | 25.3 | 10.6 | 19.9 | 44.0 | 21.6 | 0.0 | 11.2 | 29.5 | 10.9 |
| VILA | 7B | 34.5 | 39.1 | 32.3 | 46.4 | 61.9 | 35.2 | 13.5 | 29.8 | 63.5 | 21.9 | 0.0 | 11.3 | 39.6 | 11.1 |
| LLaVA-1.6-7B | 7B | 27.3 | 38.5 | 27.3 | 40.8 | 46.3 | 33.0 | 11.8 | 25.6 | 56.5 | 15.9 | 0.0 | 10.5 | 34.1 | 8.8 |
| Mantis | 7B | 43.0 | 45.1 | 34.0 | 54.5 | 68.8 | 53.2 | 17.3 | 27.2 | 61.0 | 85.3 | 0.0 | 11.8 | 44.9 | 32.4 |
| Open flamingo | 9B | 16.2 | 22.1 | 17.3 | 21.8 | 25.5 | 20.0 | 3.3 | 25.3 | 28.5 | 19.4 | 0.0 | 10.8 | 20.0 | 10.1 |
| LLaVA-1.5-13B | 13B | 33.5 | 44.1 | 29.5 | 52.4 | 64.8 | 39.0 | 13.7 | 36.0 | 64.0 | 20.0 | 0.0 | 10.0 | 41.9 | 10.0 |
| LLaVA-1.6-13B | 13B | 32.0 | 44.5 | 29.5 | 43.4 | 55.4 | 30.3 | 10.0 | 27.8 | 64.0 | 17.8 | 0.0 | 10.3 | 37.4 | 9.4 |
| *Open-source MLLMs (Video models)* | | | | | | | | | | | | | | | |
| Video-LLaMA-2 | 7B | 12.5 | 21.5 | 18.5 | 11.6 | 15.6 | 10.3 | 4.8 | 3.7 | 4.0 | 25.9 | 0.0 | 10.2 | 11.4 | 12.0 |
| Valley | 7B | 21.3 | 31.9 | 24.3 | 26.1 | 24.0 | 23.8 | 13.9 | 7.8 | 26.5 | 10.5 | 5.3 | 0.0 | 22.2 | 5.3 |
| VideoChat2 | 7B | 14.3 | 35.5 | 25.8 | 26.4 | 44.1 | 19.7 | 12.6 | 26.2 | 25.0 | 14.1 | 0.0 | 9.2 | 25.5 | 7.7 |
| LLaMA-VID | 7B | 25.8 | 33.3 | 25.8 | 35.6 | 44.9 | 24.0 | 12.0 | 28.8 | 38.0 | 51.3 | 0.0 | 10.7 | 29.8 | 20.6 |
| LWM | 7B | 1.3 | 10.4 | 0.8 | 3.1 | 8.4 | 10.3 | 9.9 | 6.8 | 7.0 | 43.1 | 0.0 | 0.2 | 6.4 | 14.4 |

evaluation (average: 51.2% v.s. 12.3%, max: 60.5% v.s. 32.4%). (2) **In comparison to the results on the multi-image set, the performance of proprietary models declined**, except for Gemini 1.0, which is limited by the number of images uploaded on the multi-image set. The potential reason is that to maintain performance on the combined image set, models need to possess high-resolution vision models that can effectively distinguish multiple images combined together. For instance, Gemini 1.0 adjusts images of excessively large resolution to a size of $3072 \times 3072$. On the other hand, GPT-4V and Claude 3 Opus resize the images to dimensions of $768 \times 768$ and $1568 \times 1568$, respectively. The lower resolution input of GPT-4V and Claude 3 Opus, in comparison to Gemini 1.0, could potentially be a factor for their diminished performance. (3) **Compared to the results on the multi-image set, the performance of some open-source models with short contexts improved**, such as ALLaVA-Longer (from 24.7% to 26.9%) and MiniGPT-v2 (from 17.8% to 29.5%). The possible reason is that these models have only been trained on single images and cannot effectively generalize to multi-image scenarios. Combining images can effectively alleviate this issue.

## D.2 Inter-task Correlation in MILEBENCH

To investigate the multi-task characteristics of the realistic evaluation in our MILEBENCH, we analyzed the performance of all models across the nine tasks within this evaluation and calculated pairwise correlations between different tasks, as shown in Figure 13. (1) We found that, aside from Task S-3 (Visual Relation Inference), tasks within the same category (either temporal multi-image or semantic multi-image) exhibited high correlation. Task S-3, being a challenging one, showed little variation in scores across models. (2) We also noted that Task T-3 (Visual Navigation and Spatial Localization) demonstrated lower correlation with other tasks, possibly due to its requirement of unique cognitive skills such as understanding the world from a first-person perspective. These observations suggest that **the realistic evaluation of MILEBENCH encompasses a broad range of task types, offering a more comprehensive assessment in the context of multi-image long-context scenarios**.

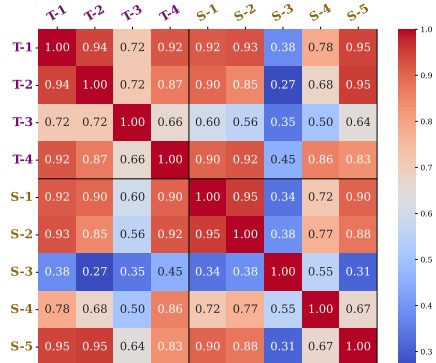

Figure 13: **Spearman correlation between each pair of tasks in realistic evaluation.**

### D.3 Risk of Data Contamination for MILEBENCH

Considering MILEBENCH's use of public datasets, there's a **potential risk of data contamination**. Our investigation, which involved excluding models trained solely on single-image tasks, selected four open-source models—Qwen-VL-Chat, Cheetor, Open Flamingo, and VILA—and one closed-source model, GPT-4o (details in Appendix C.1) We referred to Wei et al. (2023) and constructed an **Adversarial (ADV) Set** with shuffled options and paraphrased reference answers and evaluated the difference between original and ADV results.

| Model | Regular | ADV | Δ ↓ |
|---|---|---|---|
| Cheetor | 25.4 | 25.3 | 0.1 |
| VILA | 44.4 | 43.6 | 0.8 |
| Qwen-VL-Chat | 39.1 | 38.2 | 0.9 |
| Open flamingo | 27.4 | 26.2 | 1.2 |
| GPT-4o | 60.3 | 57.3 | 3.0 |

Table 10: Contamination Detection. We present Regular (result on MILEBENCH), ADV (result on the ADV set) and their difference Δ.

The results (Table 10) indicate a negligible performance drop, ranging from 0.1% to 1.2% for the open-source models and 3% for the closed-source model, **suggesting a minimal likelihood that these models were trained on our dataset**.

Table 11: **Experiment Result on Temporal Multi-image Tasks.** AL: Action Localization. AP: Action Prediction. AS: Action Sequence. CO: Character Order. CI: Counterfactual Inference. EN: Egocentric Navigation. MA: Moving Attribute. MD: Moving Direction. OE: Object Existence. OI: Object Interaction. OS: Object Shuffle. ST: Scene Transition. SC: State Change.

| Model | AL | AP | AS | CO | CI | EN | MA | MD | OE | OI | OS | ST | SC |
|---|---|---|---|---|---|---|---|---|---|---|---|---|---|
| *Random* | 25.0 | 25.0 | 25.0 | 33.3 | 34.8 | 25.0 | 36.0 | 25.0 | 33.3 | 25.0 | 33.3 | 25.0 | 33.3 |
| *Closed-source MLLMs* | | | | | | | | | | | | | |
| GPT-4V | 36.5 | 64.0 | 54.5 | 73.0 | 32.0 | 28.5 | 51.0 | 16.0 | 48.5 | 66.0 | 35.0 | 83.0 | 46.0 |
| GPT-4o | 34.5 | 73.5 | 77.5 | 86.5 | 46.0 | 44.5 | 58.0 | 31.5 | 52.0 | 79.5 | 37.5 | 86.5 | 54.0 |
| Gemini 1.0 | 30.0 | 48.0 | 46.5 | 39.0 | 41.0 | 28.5 | 48.5 | 26.0 | 53.0 | 50.5 | 32.0 | 84.5 | 42.0 |
| Gemini 1.5 | 40.0 | 63.5 | 61.5 | 75.0 | 33.5 | 32.5 | 60.5 | 35.5 | 54.0 | 68.0 | 35.5 | 79.0 | 41.5 |
| Claude 3 Opus | 32.5 | 41.5 | 39.0 | 56.5 | 30.5 | 24.5 | 46.5 | 17.5 | 45.5 | 47.5 | 29.0 | 71.5 | 36.0 |
| *Open-source MLLMs (Image models)* | | | | | | | | | | | | | |
| ALLaVA-Longer | 19.0 | 23.5 | 20.0 | 25.5 | 23.5 | 25.5 | 29.0 | 30.5 | 47.0 | 21.5 | 32.0 | 38.5 | 37.5 |
| Yi-VL | 31.5 | 22.0 | 27.0 | 40.0 | 34.0 | 33.5 | 32.5 | 29.5 | 45.5 | 28.5 | 33.0 | 51.0 | 39.5 |
| Cheetor | 25.5 | 24.5 | 22.5 | 40.0 | 23.0 | 24.5 | 30.5 | 12.0 | 11.5 | 21.5 | 29.5 | 9.0 | 40.0 |
| Qwen-VL-Chat | 32.5 | 34.0 | 37.5 | 38.0 | 30.5 | 22.0 | 42.5 | 22.5 | 41.0 | 40.5 | 36.0 | 71.5 | 38.5 |
| LLaVA-1.5-7B | 24.0 | 48.5 | 38.5 | 26.0 | 28.5 | 31.5 | 47.0 | 32.0 | 53.0 | 47.5 | 33.5 | 71.5 | 30.0 |
| MiniGPT-v2 | 9.5 | 6.5 | 11.5 | 17.5 | 39.0 | 15.0 | 20.5 | 17.0 | 14.5 | 16.5 | 6.0 | 5.5 | 19.0 |
| VILA | 23.0 | 53.0 | 45.0 | 48.5 | 38.5 | 35.0 | 54.0 | 35.5 | 54.5 | 51.5 | 36.5 | 78.0 | 32.0 |
| LLaVA-1.6-7B | 25.5 | 50.0 | 41.0 | 23.0 | 17.0 | 28.5 | 41.0 | 28.5 | 51.5 | 46.5 | 38.5 | 58.5 | 33.5 |
| Mantis | 25.0 | 60.5 | 49.0 | 41.5 | 34.0 | 20.5 | 55.5 | 30.0 | 55.5 | 57.0 | 31.5 | 66.5 | 53.5 |
| Open flamingo | 26.5 | 25.0 | 24.0 | 35.5 | 38.0 | 25.0 | 31.5 | 27.5 | 30.0 | 29.5 | 39.0 | 33.0 | 34.0 |
| LLaVA-1.5-13B | 25.5 | 45.5 | 42.5 | 43.0 | 36.0 | 26.0 | 48.5 | 37.0 | 46.0 | 45.0 | 42.5 | 70.5 | 36.0 |
| LLaVA-1.6-13B | 27.0 | 34.0 | 31.5 | 29.5 | 20.0 | 23.5 | 49.0 | 23.0 | 52.0 | 50.0 | 39.0 | 48.5 | 34.0 |
| *Open-source MLLMs (Video models)* | | | | | | | | | | | | | |
| Video-LLaMA-2 | 1.5 | 0.0 | 0.0 | 9.5 | 8.5 | 2.0 | 3.5 | 0.5 | 5.5 | 1.5 | 5.0 | 0.0 | 9.5 |
| Valley | 16.0 | 15.0 | 20.0 | 32.5 | 28.0 | 12.0 | 33.5 | 13.0 | 39.5 | 30.0 | 16.0 | 17.5 | 30.0 |
| VideoChat2 | 6.5 | 16.0 | 10.0 | 16.5 | 25.0 | 17.5 | 44.0 | 8.0 | 37.5 | 11.0 | 16.0 | 10.0 | 12.0 |
| LLaMA-VID | 27.5 | 29.0 | 22.0 | 42.0 | 39.5 | 25.0 | 46.0 | 28.0 | 30.0 | 34.0 | 39.0 | 49.0 | 31.5 |
| LWM | 0.5 | 1.0 | 0.0 | 12.0 | 2.5 | 0.5 | 13.5 | 4.5 | 40.0 | 5.0 | 0.0 | 0.5 | 10.5 |

Table 12: **Experiment Result on Semantic Multi-image Tasks.** CED: Conversational Embodied Dialogue. VCC1: Visual Change Captioning (CLEVR-Change). DQ: Document QA. VRE: Visual Relationship Expressing. MD: Multimodal Dialogue. CMQ: Complex Multimodal QA. SU: Space Understanding. OQ: OCR QA. SQ: Slide QA. VCC2: Visual Change Captioning (Spot-the-Diff). TQ: Textbook QA. WQ: Webpage QA. LTIQ: Long Text with Images QA.

| Model | CED | VCC1 | DQ | VRE | MD | CMQ | SU | OQ | SQ | VCC2 | TQ | WQ | LTIQ |
|---|---|---|---|---|---|---|---|---|---|---|---|---|---|
| *Random* | 0.0 | 0.0 | 25.0 | 0.0 | 0.0 | 25.3 | 25.3 | 23.0 | 25.7 | 0.0 | 25.0 | 25.3 | 25.0 |
| *Closed-source MLLMs* | | | | | | | | | | | | | |
| GPT-4V | 15.4 | 15.5 | 92.5 | 5.1 | 69.5 | 80.0 | 81.0 | 57.5 | 82.0 | 12.7 | 76.5 | 79.0 | 95.5 |
| GPT-4o | 24.4 | 20.3 | 96.0 | 7.6 | 70.0 | 85.0 | 86.5 | 64.0 | 90.0 | 18.1 | 80.0 | 79.5 | 96.5 |
| Gemini 1.0 | 41.1 | 21.1 | 75.5 | 9.6 | 54.5 | 76.0 | 73.0 | 52.5 | 66.0 | 19.9 | 66.0 | 67.5 | 88.0 |
| Gemini 1.5 | 26.9 | 21.3 | 88.5 | 12.8 | 62.5 | 71.5 | 82.5 | 52.0 | 77.5 | 19.4 | 74.0 | 71.5 | 78.5 |
| Claude 3 Opus | 19.6 | 19.7 | 81.5 | 8.7 | 68.5 | 56.0 | 58.5 | 31.0 | 66.5 | 11.7 | 70.0 | 61.5 | 71.5 |
| *Open-source MLLMs (Image models)* | | | | | | | | | | | | | |
| ALLaVA-Longer | 10.4 | 15.8 | 28.0 | 5.7 | 25.0 | 27.0 | 25.5 | 2.5 | 31.5 | 14.8 | 33.5 | 49.5 | 25.5 |
| Yi-VL | 30.0 | 15.2 | 53.5 | 5.4 | 37.0 | 48.0 | 35.5 | 8.0 | 41.0 | 13.7 | 51.5 | 57.0 | 72.0 |
| Cheetor | 32.1 | 29.0 | 18.0 | 13.9 | 23.0 | 32.5 | 32.0 | 28.0 | 28.5 | 22.7 | 25.0 | 29.5 | 27.0 |
| Qwen-VL-Chat | 19.4 | 15.3 | 58.0 | 8.5 | 34.0 | 66.0 | 59.5 | 42.5 | 56.5 | 19.2 | 53.5 | 44.0 | 65.5 |
| LLaVA-1.5-7B | 14.9 | 16.4 | 43.5 | 1.2 | 33.5 | 66.5 | 62.5 | 10.0 | 42.5 | 16.5 | 46.5 | 58.5 | 31.5 |
| MiniGPT-v2 | 1.9 | 5.5 | 25.5 | 7.4 | 32.0 | 34.0 | 16.5 | 21.5 | 28.5 | 10.0 | 21.0 | 52.0 | 31.0 |
| VILA | 20.0 | 16.6 | 44.0 | 7.1 | 41.5 | 72.5 | 73.0 | 31.5 | 48.5 | 13.6 | 55.5 | 65.0 | 79.5 |
| LLaVA-1.6-7B | 11.8 | 13.7 | 44.0 | 5.9 | 42.0 | 60.0 | 69.5 | 32.0 | 45.0 | 10.2 | 49.0 | 38.0 | 59.5 |
| Mantis | 26.3 | 17.6 | 53.0 | 8.7 | 37.5 | 74.5 | 78.0 | 61.0 | 52.5 | 26.0 | 39.5 | 62.5 | 86.5 |
| Open flamingo | 23.9 | 17.6 | 26.5 | 13.3 | 41.5 | 29.0 | 28.5 | 20.5 | 24.0 | 17.8 | 29.0 | 29.5 | 20.0 |
| LLaVA-1.5-13B | 19.0 | 15.9 | 46.0 | 9.7 | 40.0 | 74.0 | 59.0 | 50.5 | 51.0 | 15.7 | 55.0 | 65.0 | 73.5 |
| LLaVA-1.6-13B | 12.7 | 12.8 | 41.0 | 5.7 | 38.5 | 69.5 | 72.5 | 13.5 | 47.5 | 10.1 | 53.5 | 45.0 | 68.5 |
| *Open-source MLLMs (Video models)* | | | | | | | | | | | | | |
| Video-LLaMA-2 | 7.2 | 5.0 | 2.5 | 4.9 | 0.0 | 7.0 | 10.0 | 5.5 | 2.0 | 4.2 | 11.0 | 0.5 | 27.5 |
| Valley | 20.7 | 9.1 | 19.0 | 3.7 | 33.0 | 24.5 | 31.0 | 21.5 | 15.5 | 9.9 | 20.5 | 21.5 | 8.4 |
| VideoChat2 | 13.0 | 22.6 | 30.5 | 7.6 | 35.0 | 53.5 | 31.5 | 7.0 | 28.0 | 12.9 | 31.0 | 33.5 | 50.5 |
| LLaMA-VID | 19.9 | 14.6 | 35.0 | 6.5 | 36.0 | 37.5 | 46.5 | 14.5 | 28.5 | 13.9 | 47.0 | 37.0 | 52.5 |
| LWM | 13.3 | 14.9 | 13.5 | 6.5 | 3.5 | 17.5 | 15.0 | 17.5 | 11.0 | 10.9 | 9.0 | 1.5 | 1.0 |

Table 13: **Experiment Result on Temporal Multi-image Tasks of Combined-image Set.** AL: Action Localization. AP: Action Prediction. AS: Action Sequence. CO: Character Order. CI: Counterfactual Inference. EN: Egocentric Navigation. MA: Moving Attribute. MD: Moving Direction. OE: Object Existence. OI: Object Interaction. OS: Object Shuffle. ST: Scene Transition. SC: State Change.

| Model | AL | AP | AS | CO | CI | EN | MA | MD | OE | OI | OS | ST | SC |
|---|---|---|---|---|---|---|---|---|---|---|---|---|---|
| *Random* | 25.0 | 25.0 | 25.0 | 33.3 | 34.8 | 25.0 | 36.0 | 25.0 | 33.3 | 25.0 | 33.3 | 25.0 | 33.3 |
| *Closed-source MLLMs* | | | | | | | | | | | | | |
| GPT-4V | 38.0 | 48.0 | 36.5 | 48.0 | 36.0 | 26.5 | 47.0 | 22.5 | 58.0 | 42.0 | 23.5 | 75.0 | 39.0 |
| Gemini 1.0 | 27.0 | 31.5 | 38.5 | 45.0 | 37.5 | 35.5 | 45.5 | 26.5 | 56.5 | 45.0 | 36.0 | 81.0 | 41.0 |
| Claude 3 Opus | 35.5 | 28.0 | 34.0 | 54.5 | 34.0 | 24.5 | 43.5 | 15.5 | 44.0 | 35.0 | 32.5 | 73.5 | 38.0 |
| *Open-source MLLMs (Image models)* | | | | | | | | | | | | | |
| ALLaVA-Longer | 22.5 | 21.5 | 24.0 | 34.0 | 27.5 | 32.5 | 28.0 | 28.5 | 46.0 | 23.0 | 29.5 | 49.5 | 35.5 |
| Yi-VL | 27.5 | 25.5 | 27.5 | 41.0 | 36.5 | 36.5 | 31.5 | 20.5 | 46.5 | 26.5 | 32.5 | 60.5 | 39.0 |
| Cheetor | 31.0 | 21.0 | 17.5 | 27.5 | 32.5 | 30.0 | 23.5 | 41.0 | 10.5 | 26.5 | 23.5 | 26.5 | 12.5 |
| Qwen-VL-Chat | 28.5 | 24.0 | 30.0 | 35.5 | 25.0 | 27.5 | 46.0 | 20.0 | 47.0 | 25.5 | 30.0 | 65.5 | 36.0 |
| LLaVA-1.5-7B | 20.0 | 38.0 | 32.0 | 33.0 | 28.5 | 28.0 | 48.5 | 32.5 | 38.0 | 37.0 | 29.0 | 71.5 | 34.0 |
| MiniGPT-v2 | 29.5 | 25.0 | 25.0 | 32.5 | 38.0 | 22.0 | 26.0 | 28.5 | 31.5 | 32.5 | 39.0 | 36.5 | 46.0 |
| VILA | 63.5 | 28.5 | 35.5 | 39.5 | 37.0 | 35.0 | 31.5 | 46.5 | 33.0 | 51.0 | 29.5 | 29.5 | 73.5 |
| LLaVA-1.6-7B | 27.0 | 29.0 | 26.0 | 37.0 | 39.0 | 27.0 | 42.5 | 27.5 | 40.5 | 32.0 | 39.0 | 50.5 | 36.5 |
| Mantis | 33.0 | 47.0 | 49.0 | 44.5 | 39.0 | 37.5 | 50.0 | 30.5 | 50.0 | 48.5 | 32.0 | 85.0 | 49.5 |
| Open flamingo | 15.0 | 17.5 | 16.0 | 34.0 | 18.0 | 24.0 | 15.5 | 10.5 | 22.0 | 20.0 | 31.0 | 31.0 | 4.0 |
| LLaVA-1.5-13B | 31.0 | 33.0 | 36.5 | 46.5 | 38.0 | 27.0 | 51.5 | 32.0 | 45.5 | 37.5 | 42.0 | 79.5 | 45.5 |
| LLaVA-1.6-13B | 35.5 | 25.0 | 35.5 | 35.0 | 38.5 | 33.0 | 55.5 | 26.0 | 52.5 | 37.0 | 33.0 | 58.0 | 42.0 |
| *Open-source MLLMs (Video models)* | | | | | | | | | | | | | |
| Video-LLaMA-2 | 20.5 | 16.0 | 1.0 | 16.0 | 1.0 | 14.5 | 8.0 | 22.5 | 42.5 | 13.5 | 22.0 | 6.5 | 23.0 |
| Valley | 17.5 | 25.5 | 21.0 | 30.0 | 29.5 | 23.5 | 32.0 | 25.0 | 48.5 | 32.5 | 14.5 | 19.0 | 26.0 |
| VideoChat2 | 17.5 | 19.5 | 6.0 | 29.0 | 25.0 | 33.0 | 40.0 | 18.5 | 47.5 | 26.0 | 28.5 | 24.5 | 27.0 |
| LLaMA-VID | 29.5 | 27.0 | 21.0 | 39.5 | 36.0 | 23.5 | 33.0 | 28.0 | 30.0 | 31.0 | 39.0 | 31.5 | 35.5 |
| LWM | 0.5 | 0.5 | 3.0 | 8.5 | 0.0 | 1.0 | 9.5 | 0.5 | 5.0 | 2.5 | 24.5 | 1.0 | 3.0 |

Table 14: **Experiment Result on Semantic Multi-image Tasks of Combined-image Set.** CED: Conversational Embodied Dialogue. VCC1: Visual Change Captioning (CLEVR-Change). DQ: Document QA. VRE: Visual Relationship Expressing. MD: Multimodal Dialogue. CMQ: Complex Multimodal QA. SU: Space Understanding. OQ: OCR QA. SQ: Slide QA. VCC2: Visual Change Captioning (Spot-the-Diff). TQ: Textbook QA. WQ: Webpage QA. LTIQ: Long Text with Images QA.

| Model | CED | VCC1 | DQ | VRE | MD | CMQ | SU | OQ | SQ | VCC2 | TQ | WQ | LTIQ |
|---|---|---|---|---|---|---|---|---|---|---|---|---|---|
| *Random* | 0.0 | 0.0 | 25.0 | 0.0 | 0.0 | 25.3 | 25.3 | 23.0 | 25.7 | 0.0 | 25.0 | 25.3 | 25.0 |
| *Closed-source MLLMs* | | | | | | | | | | | | | |
| GPT-4V | 11.6 | 17.4 | 60.0 | 5.1 | 58.5 | 70.0 | 70.5 | 56.0 | 66.5 | 14.1 | 68.0 | 72.5 | 94.0 |
| Gemini 1.0 | 41.5 | 20.3 | 88.0 | 10.3 | 48.0 | 60.0 | 70.0 | 56.0 | 68.0 | 17.5 | 61.5 | 68.0 | 91.0 |
| Claude 3 Opus | 18.0 | 18.4 | 69.0 | 9.1 | 55.0 | 57.5 | 51.0 | 36.5 | 68.0 | 15.7 | 66.5 | 62.5 | 74.5 |
| *Open-source MLLMs (Image models)* | | | | | | | | | | | | | |
| ALLaVA-Longer | 11.7 | 15.5 | 30.0 | 4.7 | 38.5 | 30.0 | 29.5 | 3.0 | 30.5 | 11.2 | 41.5 | 52.0 | 12.5 |
| Yi-VL | 22.8 | 12.0 | 51.5 | 5.4 | 38.5 | 49.0 | 42.5 | 7.0 | 42.0 | 12.3 | 54.5 | 57.5 | 70.0 |
| Cheetor | 31.7 | 29.8 | 23.5 | 15.5 | 22.0 | 32.5 | 31.0 | 25.0 | 29.5 | 23.2 | 34.5 | 26.5 | 24.5 |
| Qwen-VL-Chat | 20.4 | 11.1 | 38.5 | 6.6 | 39.5 | 60.5 | 53.0 | 44.0 | 52.0 | 11.0 | 49.5 | 49.5 | 61.0 |
| LLaVA-1.5-7B | 19.8 | 16.7 | 39.0 | 8.6 | 41.0 | 62.5 | 65.5 | 4.5 | 43.5 | 21.2 | 40.5 | 59.5 | 52.5 |
| MiniGPT-v2 | 5.7 | 9.6 | 28.5 | 10.6 | 34.0 | 42.5 | 44.0 | 18.0 | 29.5 | 11.6 | 38.0 | 48.5 | 43.0 |
| VILA | 20.0 | 16.9 | 40.0 | 7.5 | 39.5 | 64.5 | 63.5 | 21.5 | 44.0 | 16.1 | 48.5 | 69.0 | 65.5 |
| LLaVA-1.6-7B | 13.8 | 16.8 | 35.0 | 5.5 | 37.5 | 52.0 | 56.5 | 21.5 | 42.5 | 13.2 | 48.5 | 37.5 | 47.0 |
| Mantis | 14.9 | 19.5 | 46.0 | 8.6 | 39.5 | 71.0 | 61.0 | 62.5 | 51.0 | 23.7 | 54.0 | 62.0 | 88.0 |
| Open flamingo | 12.1 | 1.4 | 14.0 | 6.3 | 38.5 | 31.5 | 28.5 | 21.0 | 25.0 | 2.2 | 27.5 | 25.0 | 18.0 |
| LLaVA-1.5-13B | 30.6 | 15.7 | 42.5 | 5.9 | 41.5 | 70.5 | 64.0 | 27.0 | 47.5 | 19.7 | 51.5 | 68.5 | 68.5 |
| LLaVA-1.6-13B | 13.5 | 15.0 | 44.5 | 4.9 | 42.0 | 61.0 | 64.0 | 3.0 | 43.5 | 10.1 | 54.0 | 55.0 | 51.5 |
| *Open-source MLLMs (Video models)* | | | | | | | | | | | | | |
| Video-LLaMA-2 | 5.4 | 7.5 | 2.5 | 5.5 | 2.0 | 13.5 | 4.0 | 15.0 | 13.5 | 1.6 | 9.5 | 11.5 | 28.0 |
| Valley | 9.3 | 9.8 | 17.5 | 4.3 | 6.3 | 34.5 | 26.5 | 34.5 | 19.5 | 27.5 | 13.5 | 25.0 | 23.0 |
| VideoChat2 | 19.4 | 18.4 | 29.0 | 6.6 | 33.0 | 51.5 | 25.0 | 4.0 | 26.0 | 13.0 | 36.0 | 45.0 | 44.0 |
| LLaMA-VID | 19.2 | 16.6 | 31.0 | 5.9 | 38.5 | 39.5 | 38.0 | 12.5 | 28.5 | 13.5 | 44.5 | 35.5 | 60.0 |
| LWM | 8.1 | 12.8 | 10.0 | 6.4 | 5.5 | 11.5 | 7.0 | 8.5 | 12.5 | 10.5 | 13.5 | 2.5 | 6.0 |

