# OpenReview forum: "MileBench: Benchmarking MLLMs in Long Context"
_colmweb.org/COLM/2024/Conference — COLM_

### Official Review · Reviewer_u8ee · 2024-05-10

**Rating:** 7
**Confidence:** 3
**Ethics Flag:** 1

**Summary:**

The paper presents a new benchmark dataset for multimodal (image+text) LLMs, called MileBench.

The novelty of this benchmark is in its focus - testing multimodal models in long contexts with many images. The collected dataset is a subset of existing datasets and synthetically generated examples. The examples from the existing datasets were picked in a way which agrees with the benchmark's focus. The resulting dataset consists of around 6k examples categorized in 12 different tasks.

The authors also measured the performance of various models on this benchmark and report the results. Notably, the closed-source models emerged as clear winners on most of the tasks.

Overall, the paper is well written and presents a relevant and potentially useful contribution.

**Questions To Authors:**

* Have you tried using different metrics than ROUGE-L? I think for open ended generation tasks it might make more sense to use a LLM-based metric instead. I realize that selecting a good metric is secondary to this paper's focus, but by introducing a benchmark like this, you basically dictate which metric to use if someone else decides to use the benchmark to compare with other models.

* Given that the collected benchmark is a subset of existing datasets, is it possible that the success of the closed-source models is determined by a data leakage? (https://aclanthology.org/2024.eacl-long.5/) Why are these models omitted in the analysis in section 4.3.3?

Some minor suggestions:
For better clarity, Table 2 should indicate which numbers are accuracy and which are Rouge-L scores.

**Reasons To Accept:**

* A new benchmark dataset which might be useful for multimodal LLM research.
* Includes baseline scores for many publicly available models, making comparison using this benchmark more accessible.

**Reasons To Reject:**

* It is not very clear which datasets have been used for the data collection. References are given to the tasks, so I am assuming the data comes from there, but this should be listed explicitly, perhaps in the appendix.
* Inadequate attention given to possible data contamination in discussion about the results (see my question below).

---

> ### Author Rebuttal · Authors · 2024-05-31
>
> We appreciate the reviewer's recognition of the benchmark’s usefulness and the thoroughness of our experiments. In response to the concerns raised, we provide the following clarifications:
>
> > Q1: Clarification on Data Sources
>
> Thank you for your query regarding the datasets used for data collection. We have listed the datasets in Table 5 of Appendix B.1. We agree that it is important to emphasize and highlight the data sources in the final version of the paper for better clarity.
>
> > Q2: Choice of Metrics
>
> Regarding the choice of metrics, we have adhered to the metrics used by the original datasets. For the datasets we constructed ourselves, we employed standard metrics for multiple-choice questions and extractive question types. Due to the high cost of using LLM-based metrics, the instability of the evaluator model versions, and the self-enhancement bias present in the evaluation models, we did not consider using LLM-based metrics for evaluation in the current version. Of course, in future versions, with the addition of new tasks, LLM-based metrics will be an effective supplement to the existing metrics, and we will consider adding LLM-based metrics as appropriate.
>
> > Q3: Evaluation of Closed-Source Models
>
> Due to cost considerations, we did not initially evaluate the closed-source models. However, based on your suggestion, we agree that it is necessary to include these evaluations. We conduct the necessary experiment of GPT-4o and include the result:
>
> | Model         | Regular | ADV   | $\Delta \downarrow$ |
> |-------------------|---------|-------|----------------------|
> | GPT-4o            | 60.3    | 57.3  | 3.0                  |
>
> The results show that GPT-4o has certain data leakage issues, with an overall score drop of about 3 points. However, overall, the data leakage situation is not serious, and the score drop caused by changing the options did not significantly affect the ranking order of the model.
>
> > Q4: Clarity in Table 2
>
> We will modify Table 2 to clearly indicate which numbers represent accuracy and which represent ROUGE-L scores for better clarity.

---

> > ### Comment · Reviewer_u8ee · 2024-06-04
> > **Response**
> >
> > Thanks for pointing out that the data sources are actually already listed in the appendix - it's my bad I didn't notice previously. Since I presented is originally as a reason to reject, I am raising the score to 7.
> >
> > I acknowledge the rest of the answers, however they don't particularly ease my concerns about insufficient data leakage discussion in the paper.

---

### Official Review · Reviewer_FSLA · 2024-05-10

**Rating:** 6
**Confidence:** 4
**Ethics Flag:** 1

**Summary:**

This paper focuses on evaluating the long-context understanding capabilities of multimodal large language models. The authors introduce MILEBENCH, a comprehensive benchmark encompassing multiple dimensions of multimodal long-context understanding, such as Temporal Multi-image Understanding, Semantic Multi-image Understanding, Needle in a Haystack, and Image Retrieval. Additionally, the paper provides a brief evaluation of various multimodal models, ranging from closed-source to open-source, trained on images and videos, presenting a comparative analysis of their performance.

**Questions To Authors:**

Refer to the Reasons To Reject.

**Reasons To Accept:**

1. Long-context capabilities, especially in the domain of multi-image, long-context understanding, are crucial functionalities of multimodal large language models, yet they are absent in many current models. A comprehensive benchmark can facilitate the development of these capabilities.
2. This paper provides an interesting and significant insight: numerous models that excel in common benchmarks show weak performance in multi-images scenario.
3. This paper is well organized, offering concise experiments across a range of popular multimodal large language models.

**Reasons To Reject:**

1. This paper collects data from publicly available datasets/benchmarks, yielding only 200 samples for each task. However, some of the pre-existing datasets used herein (e.g., START [1]) encompass a vast collection of video clips and annotations potentially suitable for directly assessing multi-image capabilities. It's not clear whether these datasets alone are inadequate. The authors can provide substantial evidence to justify the necessity of the proposed benchmarks.
2. Recent developments have seen certain models displaying capabilities in multiple images understanding (e.g., MM1 [2], mPLUG-Owl2 [3], MMICL [4]). These models should be included in the evaluation to provide a comprehensive understanding of their capabilities and how training strategy/training data/model designing affect the multimodal long-context understanding capabilities.
3. The experiment results indicate that some models struggle to follow instructions and perform poorly on several samples. It begs the question of whether these models inherently lack the multimodal long-context understanding or if their performance is hindered by unfamiliar patterns that might be improved with training on a few of data samples.  In essence, the authors are encouraged to delve deeper into discussing which fundamental abilities, which preventing the models from effectively tackling multimodal long-context problems, are absent.

[1] Wu, B., Yu, S., Chen, Z., Tenenbaum, J. B., & Gan, C. (2021, August). Star: A benchmark for situated reasoning in real-world videos. In Thirty-fifth conference on neural information processing systems datasets and benchmarks track (Round 2).
[2] McKinzie, B., Gan, Z., Fauconnier, J. P., Dodge, S., Zhang, B., Dufter, P., ... & Yang, Y. (2024). Mm1: Methods, analysis & insights from multimodal llm pre-training. arXiv preprint arXiv:2403.09611.
[3] Ye, Q., Xu, H., Ye, J., Yan, M., Liu, H., Qian, Q., ... & Zhou, J. (2023). mplug-owl2: Revolutionizing multi-modal large language model with modality collaboration. arXiv preprint arXiv:2311.04257.
[4] Zhao, H., Cai, Z., Si, S., Ma, X., An, K., Chen, L., ... & Chang, B. (2023). Mmicl: Empowering vision-language model with multi-modal in-context learning. arXiv preprint arXiv:2309.07915.

---

> ### Author Rebuttal · Authors · 2024-05-31
>
> We appreciate the reviewer's recognition of the significance and concise experiments of our work. In response to the concerns raised, we provide the following clarifications:
>
> > Q1: Justification for the Proposed Benchmark
>
> Our dataset differs from video benchmarks in that our benchmark is **more comprehensive** and **focuses on the overall multi-image and long-context understanding capabilities** of MLLMs, while videos are just one form and source of data.
>
> > Q2: Inclusion of Recent Models in Evaluation
>
> Thank you for your suggestions:
> 1. We have evaluated the models mPLUG-Owl2, MMICL, and Mantis (MM1 has not been open-sourced). The evaluation results are as follows:
>
> | Model         | Size | T-1 | T-2 | T-3 | T-4 | S-1 | S-2 | S-3 | S-4 | S-5 | N-1 | N-2 | I-1 | Overall Real. | Overall Diag. |
> |---------------|------|-----|-----|-----|-----|-----|-----|-----|-----|-----|-----|-----|-----|---------------|---------------|
> | mPLUG-Owl2    | 7B   | 49.0| 45.3| 23.8| 46.6| 57.5| 57.3| 16.3| 25.9| 64.5| 18.4| 9.4 | 10.7| 42.9          | 12.8          |
> | MMICL         | 12B  | 59.0| 48.1| 24.5| 49.9| 66.6| 37.0| 11.9| 34.1| 69.0| 50.3| 0.0 | 37.2| 44.5          | 29.2          |
> | Mantis        | 8B   | 54.8| 49.9| 25.3| 48.9| 65.8| 55.5| 17.4| 31.9| 78.0| 27.5| 0.0 | 32.2| 47.5          | 19.9          |
>
> 2. Due to the variety of strategies used by different models, we **cannot** conclusively determine that a single strategy is beneficial for multi-image understanding. However, we can still make some general observations based on models with similar strategies and append to the final version, such as:
>     1. **Training with multi-image data can improve performance**: Qwen-VL-Chat, VILA, Mantis.
>     2. **LLMs with long-context capabilities aid in multimodal long-context understanding**: Qwen-VL-Chat, Mantis.
>
> > Q3: Fundamental Abilities for Tackling Multimodal Long-Context Problems
>
> Mantis is an LLaMA3-based model trained on a large multi-image dataset. Our analysis of Mantis' results suggests that **training with multi-image samples can enhance a model's understanding of multimodal long-context patterns**. However, **this alone is not sufficient to achieve promising performance**, especially in the "Needle in a Haystack" tests where visual perception is also critical. In the final version, we will include more model results and detailed analysis and discussion.

---

> > ### Comment · Reviewer_FSLA · 2024-06-04
> > **Official Comment for Response**
> >
> > Thanks for the response, I would like to keep my score.

---

### Official Review · Reviewer_DvvC · 2024-05-11

**Rating:** 6
**Confidence:** 4
**Ethics Flag:** 2

**Summary:**

The paper introduces MILEBENCH, a benchmark designed to assess the capabilities of Multimodal Large Language Models (MLLMs) in handling long-context scenarios involving multiple images and texts. This benchmark is motivated by the limitations of existing benchmarks that predominantly focus on short texts and single images, which do not adequately represent real-world multimodal tasks that require handling complex, lengthy contexts with multiple images. MILEBENCH includes two types of evaluations: diagnostic and realistic, aiming to thoroughly assess the performance of MLLMs in complex scenarios that mimic real-world applications. The benchmark tests 19 different models, highlighting that while the proprietary model GPT-4(Vision) excels, most open-source models struggle with these tasks.

**Ethics Concerns Details:**

This paper release new datasets.

**Questions To Authors:**

Please check the "Reasons To Reject"

**Reasons To Accept:**

1. The introduction of MILEBENCH addresses a significant gap in the evaluation of MLLMs by focusing on long-context multimodal scenarios, which are crucial for real-world applications but often overlooked in other benchmarks.
2. The paper does a good job of designing a variety of tasks that test different aspects of MLLMs, such as temporal and semantic understanding, across both realistic and diagnostic settings.

**Reasons To Reject:**

1. Majority of the curated benchmark comes from existing dataset, which may have data leakage and lead to unfairness and bias. Moreover, which part of the existing dataset did you extract the data from, training set, validation set, or test set?
2. While MILEBENCH incorporates a variety of tasks and sources, there is little discussion on the potential biases inherent in these sources or how they might affect the benchmark's generalizability.

---

> ### Author Rebuttal · Authors · 2024-05-31
>
> We appreciate the reviewer's recognition of the significance of our work and the comprehensiveness of our taxonomy. In response to the concerns raised, we provide the following clarifications:
>
> > Q1: Potential Data Leakage
>
> We have discussed the potential risk of data leakage in Section 4.3.3 and have confirmed that MileBench currently has **minimal risk** of such leakage. The data we extracted from other datasets is solely from their test sets.
>
> > Q2: Potential Biases in Data Sources
>
> The data sources for MileBench are either existing datasets or publicly available wiki documents. For the former, these datasets have undergone peer review and are widely used, ensuring that any bias issues are addressed by their original creators. For the data generated using wiki documents, while we have implemented measures to ensure the suitability of the content, we acknowledge the potential existence of problematic content. We are committed to continuously improving our dataset to mitigate these issues.

---

> > ### Comment · Reviewer_DvvC · 2024-06-05
> > **Thanks for the response**
> >
> > Thanks for your response. I would maintain the rating.

---

> > ### Comment · Reviewer_DvvC · 2024-06-05
> > **Thanks for the response**
> >
> > Thanks for your response. I would maintain the rating.

---

### Official Review · Reviewer_oeBf · 2024-05-13

**Rating:** 6
**Confidence:** 5
**Ethics Flag:** 1

**Summary:**

This paper presents a new benchmark, with a focus on multi-image scenarios. The benchmark mainly evaluates the long-context performance on the M-LLMs.

The paper provides a comprehensive evaluation of several existing models, and shows that long-context capability is demanding for improvements.

**Reasons To Accept:**

- The paper presents a comprehensive performance benchmarking.

- Compared to prior studies, the proposed MileBench benchmark can be suitable for evaluating multimodal long-context abilities of MLLMs.

- The evaluation taxonomy is clearly defined to evaluate two major categories: realistic and diagnostic evaluation. Each category includes a diverse set of evaluation tasks for evaluating the performance of multi-image scenarios.

- The paper offers error analysis to help understand the challenging cases in the proposed dataset.

**Reasons To Reject:**

- The testing samples of the proposed dataset are mostly manually borrowed from existing datasets. There is no reliable pipeline to further collect and scale up.

- It remains difficult to automatically evaluate these models which have free-form text outputs. The experiments only consider ROUGE-L and retrieval accuracy as the main metric. This may limit the long-context output scenarios.

---

> ### Author Rebuttal · Authors · 2024-05-31
>
> We appreciate the reviewer's positive comments on the comprehensiveness of our taxonomy and the detailed error analysis. In response to the concerns raised, we provide the following clarifications:
>
> > Q1: Dataset Composition and Metric Selection
>
> We acknowledge the reviewer's concern regarding the dataset composition. Currently, most of the data in MileBench is sourced from **existing public datasets' test sets**, with only a few examples being manually constructed. In the manually constructed data portion, we have demonstrated the potential of using such data as test cases. We aim to include a broader range of data sources and employ more rigorous methods to expand our multi-modal long-context benchmark in the next version.
>
> > Q2: The Use of ROUGE-L and Accuracy Metrics
>
> The reason we use ROUGE-L and accuracy as metrics is that we are **following the standards set by the datasets we collected**. These metrics are widely used and accepted for the task types addressed in our benchmark [1,2]. Our current tasks do not yet consider long-context outputs, but we will consider incorporating such tasks in the next version.
>
> > Reference:
> >
> > 1.Kuratov Y, Bulatov A, Anokhin P, et al. In Search of Needles in a 10M Haystack: Recurrent Memory Finds What LLMs Miss[J]. arXiv preprint arXiv:2402.10790, 2024.
> >
> > 2.Reid M, Savinov N, Teplyashin D, et al. Gemini 1.5: Unlocking multimodal understanding across millions of tokens of context[J]. arXiv preprint arXiv:2403.05530, 2024.

---

> > ### Comment · Reviewer_oeBf · 2024-06-03
> >
> > I have no further questions at this point and would like to keep my initial rating.

---

### Decision · Program_Chairs · 2024-07-10

**Decision:**

Accept

**Comment:**

The authors propose a new benchmark for testing the long-context capability of MLLMs, which involves inputs with a large number of image and text tokens. They provide extensive evaluation results of frontier MLLMs. While similar benchmarks exist for LLMs (like "needle in a haystack"), this is the first benchmark to test long-context capabilities in MLLMs. It includes both artificial and realistic scenarios.

Some reviewers raised concerns, such as:

- Some MLLMs specialized in long-context processing were missing in the evaluation.
- Missing discussion on whether the current low performance is a fundamental issue.
- Missing evaluation of closed LLMs and potential data contamination.

However, the authors provided satisfactory answers during the rebuttal.
Overall, all the reviewers were eventually satisfied, and we recommend acceptance.